# Defect engineering on $V_2O_3$ cathode for long-cycling aqueous zinc metal batteries

Kefu Zhu[1,7], Shiqiang Wei[1,7], Hongwei Shou[1,2,7], Feiran Shen[3,7], Shuangming Chen [1✉], Pengjun Zhang[1], Changda Wang[1], Yuyang Cao[1], Xin Guo[1], Mi Luo[4], Hongjun Zhang [4], Bangjiao Ye[4], Xiaojun Wu [2], Lunhua He[3,5,6✉] & Li Song [1✉]

Defect engineering is a strategy that is attracting widespread attention for the possibility of modifying battery active materials in order to improve the cycling stability of the electrodes. However, accurate investigation and quantification of the effect of the defects on the electrochemical energy storage performance of the cell are not trivial tasks. Herein, we report the quantification of vanadium-defective clusters (i.e., up to 5.7%) in the $V_2O_3$ lattice via neutron and X-ray powder diffraction measurements, positron annihilation lifetime spectroscopy, and synchrotron-based X-ray analysis. When the vanadium-defective $V_2O_3$ is employed as cathode active material in an aqueous Zn coin cell configuration, capacity retention of about 81% after 30,000 cycles at 5 A g$^{-1}$ is achieved. Density functional theory calculations indicate that the vanadium-defective clusters can provide favorable sites for reversible Zn-ion storage. Moreover, the vanadium-defective clusters allow the storage of Zn ions in $V_2O_3$, which reduces the electrostatic interaction between the host material and the multivalent ions.

[1] National Synchrotron Radiation Laboratory, CAS Center for Excellence in Nanoscience, University of Science and Technology of China, 230029 Hefei, China. [2] School of Chemistry and Material Sciences, University of Science and Technology of China, 230026 Hefei, China. [3] Spallation Neutron Source Science Center, 523803 Dongguan, China. [4] State Key Laboratory of Particle Detection and Electronics & Hefei National Laboratory for Physical Sciences at the Microscale, University of Science and Technology of China, 230026 Hefei, China. [5] Beijing National Laboratory for Condensed Matter Physics, Institute of Physics, Chinese Academy of Sciences, 100190 Beijing, China. [6] Songshan Lake Materials Laboratory, 523808 Dongguan, China. [7] These authors contributed equally: Kefu Zhu, Shiqiang Wei, Hongwei Shou, Feiran Shen. ✉email: csmp@ustc.edu.cn; lhhe@iphy.ac.cn; song2012@ustc.edu.cn

With the increase of energy crisis and environmental pollution problems, it is essential to develop green and clean energy storage devices. As a bellwether in the field of energy storage, lithium-ion batteries (LIBs) are a key electrochemical energy storage device[1,2]. However, emerging worries are their limited lithium resources and hazard issues towards future large-scale applications[3–6]. Encouragingly, rechargeable aqueous zinc-ion batteries (ZIBs) have emerged as the most promising complements to LIBs owing to the high theoretical specific capacity (820 mA h g$^{-1}$), low cost, abundant resources, and environmental friendliness[7–9]. Aqueous ZIBs were first introduced by Kang et al.[10]. Afterward, Mn-based oxides[11], Prussian blue analogs[12], and V-based oxides were widely studied as potential aqueous ZIBs cathode materials. However, the structural instability of Mn-based oxides during cycling brings about poor stability and the low specific capacity (<100 mA h g$^{-1}$) of Prussian blue analogs, thus restricting its further development, while vanadium has a low price, rich crystal structure, abundant reserves, and multiple valence states, which makes its oxides have a higher specific capacity and better stability. However, significant upgrading is needed to widen its application. The search for long-cycling V-based oxides cathodes for aqueous ZIBs still remains one of the most compelling issues due to serious consequences of stability penalty caused by the aqueous system and bigger ionic radius of Zn$^{2+}$. To promote the stability of electrodes, tremendous efforts have been devoted in the past few years. One common strategy used is defect engineering, in which the strong electrostatic interaction between the host and multivalent ions with a larger charge can be efficiently reduced. This can accelerate the reaction kinetics and facilitate the reversible storage of Zn ions[13–16]. Particularly, the research of defects in oxide electrodes most used for ZIBs is desirable due to the complex composition and the dynamic process during the working process. Accurate quantification of material defects is essential to determine the metal vacancies and oxygen vacancies at the same time. However, it is extremely hard to simultaneously determine the concentration of such two vacancies. For example, Rietveld refinement of X-ray diffraction (XRD) measurements is commonly employed to quantify defect concentration. Nevertheless, light oxygen is difficult to be detected by XRD owing to its small atomic radius, compared with most metal elements. Rietveld refinement of neutron powder diffraction (NPD) measurements is also often performed to quantify defect concentration. However, the scattering factor of heavier elements is too small to be probed by the mean of NPD[17]. Consequently, the joint application of multiple spectroscopy is essential to accurately quantify defects.

In this work, we have quantified that vanadium defective-V$_2$O$_3$ (V$_d$–V$_2$O$_3$) electrode in aqueous ZIBs contains 5.7% V$_d$ clusters by Rietveld refinement with combined XRD and NPD patterns. Notably, the V$_2$O$_3$ cathode containing V$_d$ clusters can deliver long cycling stability (81% retention after 30,000 cycles at a specific current of 5 A g$^{-1}$). Accurately quantifying and identifying the effect of defects provide a better understanding to further rational design of cathodes with long stability for energy storage devices.

## Results and Discussion

### Structure and morphology characterization of V$_d$–V$_2$O$_3$.
The V$_d$–V$_2$O$_3$ cathode was designed by a hydrothermal method and an ensuing annealing process (see Methods section for details). Scanning electron microscopy (SEM, Fig. 1a) and transmission electron microscope (TEM, Fig. 1b) images show that the V$_d$–V$_2$O$_3$ is a uniform multi-layer hierarchical structure assembled by thin nanosheets with a size of around 100-200 nm. In the high-resolution TEM (HRTEM, Fig. 1c) image of the V$_d$–V$_2$O$_3$, a

lattice fringe with a layer spacing of $d = 0.27$ nm was observed, corresponding to the (104) lattice plane. The surface area and pore size were detected by the physical adsorption method. The V$_d$–V$_2$O$_3$ exhibited a large local pore size of about 22 nm (Fig. 1d inset). According to the Brunauer-Emmett-Teller (BET) method, the V$_d$–V$_2$O$_3$ displayed a high surface area of 60.34 m$^2$ g$^{-1}$ (Fig. 1d), which can provide sufficient contact between the electrode and electrolyte and shorten the Zn$^{2+}$ diffusion path time. The mesoporous structure (2–50 nm)[18] is advantageous to the insertion and extraction of Zn$^{2+}$, which can effectively improve the cycle life of the battery.

For detailed structure information, X-ray absorption fine structure (XAFS) measurements were carried out to investigate the local structure of V$_d$–V$_2$O$_3$. As illustrated in the XANES spectra of V K-edge (Fig. 2a), the absorption edge of the V$_d$–V$_2$O$_3$ is found to shift toward higher energy (Site B) compared with commercial V$_2$O$_3$ (c-V$_2$O$_3$), implying a higher average valence state in V$_d$–V$_2$O$_3$. The high-resolution X-ray photoelectron spectroscopy (XPS) measurements of V 2$p$ further show that the valence states of V in both V$_d$–V$_2$O$_3$ and c-V$_2$O$_3$ are +3 and +4 coexistence (Supplementary Fig. 1). As comparison, the ratio of V$^{4+}$ in V$_d$–V$_2$O$_3$ and c-V$_2$O$_3$ is 54.32% and 46.49% (Supplementary Table 1), indicating that the surface of V$_d$–V$_2$O$_3$ also processes a higher valence state. The identical result can be gotten from electron paramagnetic resonance (EPR) spectroscopy measurements, where the EPR signal of tetravalent vanadium has a stronger response strength[19,20] (Supplementary Fig. 2). It has been reported that the surface of V$_2$O$_3$ is vulnerable to be oxidized to V$^{4+}$, which explains the existence of V$^{4+}$ in c-V$_2$O$_3$[21,22]. But the situation in V$_d$–V$_2$O$_3$ is different, the surface of V$_d$–V$_2$O$_3$ is proved to be uniformly coated with carbon, which comes from the pyrolysis of organic substances such as urea in the raw material (Supplementary Figs. 3 and 4). Besides, Raman spectroscopy measurements further confirmed the presence of carbon in V$_d$–V$_2$O$_3$, but not in c-V$_2$O$_3$ (Supplementary Figure 5). Thermogravimetric analysis (TGA) shows that the carbon content is 22.92% (Supplementary Fig. 6). Surface coated with the carbon of V$_d$–V$_2$O$_3$ is believed to be not easily oxidized at room temperature[23], so the higher content of V$^{4+}$ in V$_d$–V$_2$O$_3$ may attribute to the existence of vanadium vacancies, which leads to a valence increase of V. The pre-edge peak in the XANES of V K-edge corresponds to the electronic transition from 1$s$ to 3$d$[24], which can promulgate the local structure symmetry. As shown in Fig. 2a (Site A), the increase of pre-edge peak intensity attributes to the decrease of local symmetry of V$_d$–V$_2$O$_3$. That is suggested that the structure of V$_d$–V$_2$O$_3$ is distorted around V atoms, owing to the absence of surrounding atoms. To further obtain the accurate coordination numbers (CN), the corresponding Fourier-transformed EXAFS is fitted as shown in Supplementary Fig. 7 and the detailed fitting results can be found in Supplementary Table 2. The results demonstrate that the CN of the V-V in V$_d$–V$_2$O$_3$ is significantly lower than that in c-V$_2$O$_3$ (2.2 vs. 4), confirming that there are vanadium vacancies in V$_d$–V$_2$O$_3$[25–27].

To further corroborate the defect situation in V$_d$–V$_2$O$_3$, we used positron annihilation lifetime spectroscopy (PALS) measurements to explore the defect type and concentration of the material[28]. Table 1 shows the PALS results of V$_d$–V$_2$O$_3$ and c-V$_2$O$_3$. The PAL spectra are well decomposed into three life components ($\tau_1$, $\tau_2$, and $\tau_3$). The shortest lifetime component ($\tau_1$) corresponds to the positron annihilation in the defect-free bulk regions and tiny vacancies. The longer lifetime component ($\tau_2$) is attributed to the positron annihilation in vacancy clusters or boundary regions. Combined with the following XRD and NPD refinement results, the $\tau_2$ of c-V$_2$O$_3$ probably originates from positron annihilation in the boundary regions[29]. The longest component ($\tau_3$) of several nanoseconds often results from the

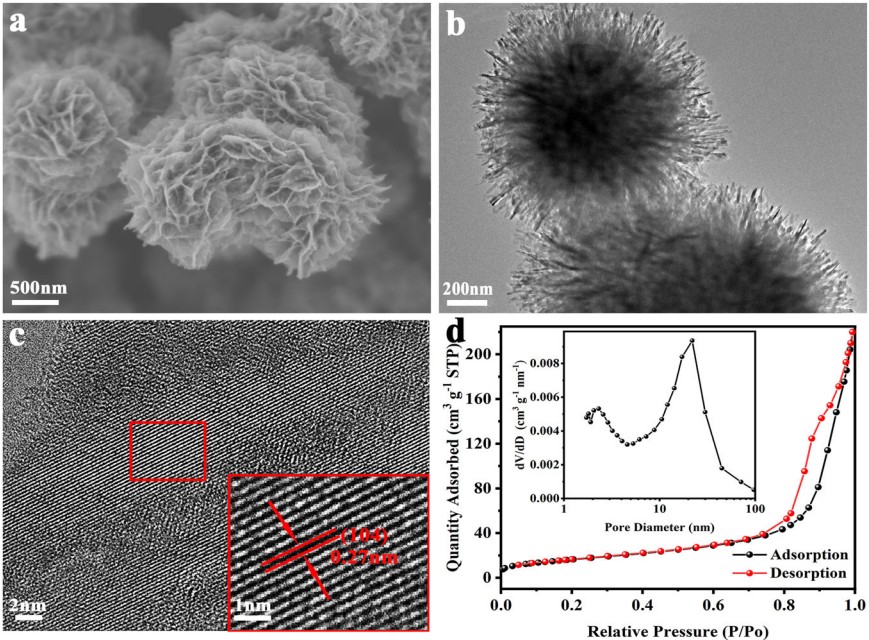

**Fig. 1 Physicochemical characterization of V$_d$–V$_2$O$_3$. a** SEM image. **b** TEM image. **c** HRTEM image. **d** Nitrogen adsorption and desorption isotherms. inset: pore size distribution.

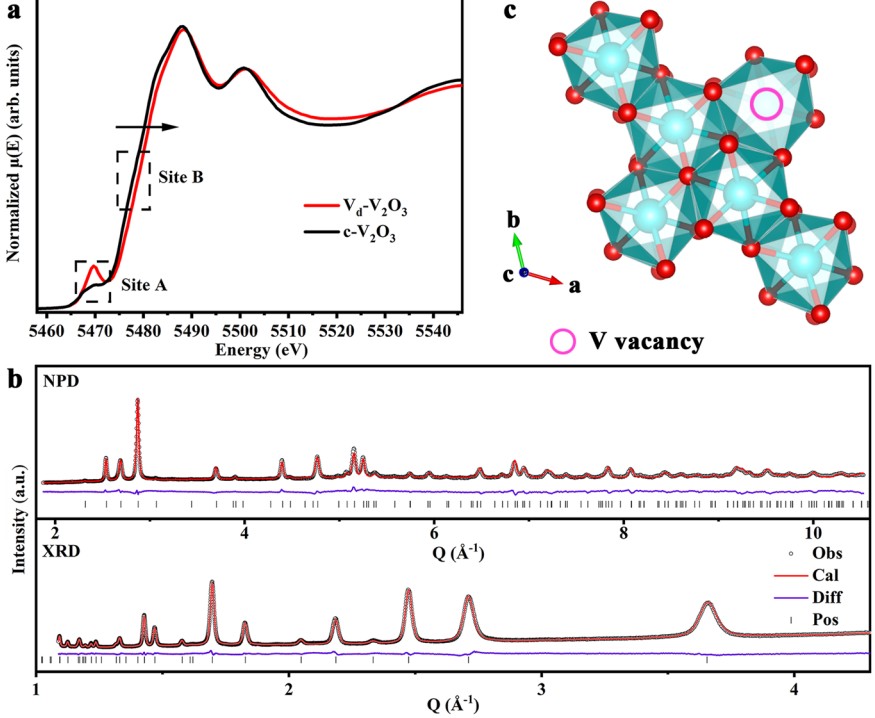

**Fig. 2 Structural characterization of V$_d$–V$_2$O$_3$. a** Normalized XANES spectra of V K-edge for V$_d$–V$_2$O$_3$ and c-V$_2$O$_3$. **b** Observed (black circle), calculated diffraction patterns (red line), their difference (purple line) and peak position (black bar) of the NPD pattern (upper part) and XRD pattern (lower part). **c** Schematic diagram of the structure of V$_d$–V$_2$O$_3$ along the c-axis. The oxygen atoms are represented by small red spheres, and the V atoms are depicted in blue.

**Table 1 Position lifetime parameters of V$_d$–V$_2$O$_3$ and c-V$_2$O$_3$.**

| Sample | $\tau_1$ (ps) | $\tau_2$ (ps) | $\tau_3$ (ns) | $I_1$ (%) | $I_2$ (%) | $I_3$ (%) |
|---|---|---|---|---|---|---|
| V$_d$–V$_2$O$_3$ | 156 | 373 | 1.559 | 17.02 | 78.01 | 4.97 |
| c-V$_2$O$_3$ | 175 | 392 | 1.610 | 57.09 | 37.94 | 4.97 |

annihilation of ortho-positronium (boundary state of a positron and an electron, spin triplet) in some large voids which is the unoccupied space among close-packed nanograins. It is not correlated with the microstructure in nanograins[30,31], thus, we could neglect the longest component. The values of $\tau_1$ (175 ps), $\tau_2$ (392 ps), and $\tau_3$ (1.610 ns) in c-V$_2$O$_3$ are very close to the previous results (171 ps, 414 ps, and 1.8 ns)[32]. For the V$_d$–V$_2$O$_3$

sample, the shortest lifetime component is lower than that of c-$V_2O_3$, indicating the significant existence of vacancy-type defects in $V_d$–$V_2O_3$, while large vacancy-type defects are not identified in c-$V_2O_3$. It is noteworthy that the intensity $I_2$ of $\tau_2$ for $V_d$–$V_2O_3$ is 78.01%, further confirming that the concentration of vacancy clusters in $V_d$–$V_2O_3$ is much higher than that in c-$V_2O_3$. The PALS results provide reliable and valuable proof of the coexistence of vanadium vacancies with a relatively high concentration in $V_d$–$V_2O_3$, and the phenomenon of defect aggregation may occur.

To accurately determine the type and concentration of defects, we combined NPD and XRD techniques. In the $V_2O_3$ cathode, the neutron scattering amplitude of the V element is just −0.0382 cm$^{-12}$, and the atomic radius of the light oxygen element is ~0.66 Å. As a result, the vanadium element in the $V_d$–$V_2O_3$ cathode is hard to be detected by NPD, while the light oxygen element is too small to be probed by XRD. However, the neutron scattering amplitude of the light oxygen element is as high as 0.5803 cm$^{-12}$, which is easy to be detected by NPD. The atomic radius of the vanadium element is 1.22 Å, which is big enough to be probed by XRD. Therefore, Rietveld refinement with combined XRD and NPD patterns was conducted to reveal the crystal structure (Fig. 2b). The refinement results (Supplementary Table 3) show that the $V_d$–$V_2O_3$ has a typical corundum-type hexagonal structure (Space group: $R$-$3c$) with lattice parameters to be $a = b = 4.9473(1)$ Å, $c = 13.9990(5)$ Å. The V and O atoms occupy the 12c (0, 0, 0.15437(6)) and 18e (0.3145(3), 0, 0.25) crystallographic positions, respectively. Moreover, the occupancy rate of vanadium atoms at 12c sites is ~94.3(1)%, while no oxygen vacancy was detected at 18e sites (Fig. 2c). In addition, the Rietveld analysis of c-$V_2O_3$ by combined XRD and NPD patterns also exhibits the crystal structure information as shown in Supplementary Fig. 8. Like $V_d$–$V_2O_3$, c-$V_2O_3$ belongs to the identical space group, with similar lattice parameters and the positions occupied by V and O atoms (Supplementary Table 3). Nevertheless, the occupation ratios of vanadium and oxygen atoms are ~100%, indicating that no vanadium vacancies and oxygen vacancies are identified in c-$V_2O_3$. To control vanadium vacancies concentration, we altered the calcination time for 0.5 h and 6 h (0.5 h-$V_d$–$V_2O_3$ and 6 h-$V_d$–$V_2O_3$) respectively. The structure information of 0.5 h-$V_d$–$V_2O_3$ and 6 h-$V_d$–$V_2O_3$ was revealed by Rietveld refinement with combined XRD and NPD patterns. The refined results are shown in Supplementary Table 3, suggesting that the structures of 0.5 h-$V_d$–$V_2O_3$ and 6 h-$V_d$–$V_2O_3$ are the same as the $V_d$–$V_2O_3$. However, the occupation ratios of 0.5 h-$V_d$–$V_2O_3$ and 6 h-$V_d$–$V_2O_3$ are 97.0(4)% and 95.3(1)%, respectively, while no oxygen vacancies are found (Supplementary Figs. 9 and 10). Accordingly, it can be concluded that the $V_d$–$V_2O_3$, 0.5 h-$V_d$–$V_2O_3$, and 6 h-$V_d$–$V_2O_3$ contain 5.7%, 3.0%, and 4.7% vanadium vacancies respectively, and no oxygen vacancies are identified, while c-$V_2O_3$ has neither vanadium vacancies nor oxygen vacancies.

**Electrochemical energy storage measurements.** To investigate the Zn$^{2+}$ storage performance of the $V_d$–$V_2O_3$ cathode, the 2032 type coin-cells were assembled using a zinc foil anode, a 3 M Zn(CF$_3$SO$_3$)$_2$ electrolyte (Supplementary Figs. 11 and 12), and a filter paper separator. As shown in Supplementary Fig. 13, the cyclic voltammetry (CV) curves are carried out at a scan rate of 0.1 mV s$^{-1}$ within a voltage window of 0.1–1.3 V (vs Zn/Zn$^{2+}$). Two pairs of redox peaks located at 1.09/0.93 V and 0.78/0.53 V are observed, which attributes to a two-step (de)intercalation process of Zn$^{2+}$. The rate performance at specific currents from

0.1 to 4.0 A g$^{-1}$ is presented in Fig. 3a, b. The reversible capacities of the Zn||$V_d$–$V_2O_3$ cell are 196, 187, 165, 147, 138, 125, 117 and 113 mA h g$^{-1}$ at the specific currents of 0.1, 0.3, 0.5, 0.8, 1.0, 2.0, 3.0, and 4.0 A g$^{-1}$, respectively. When specific current returns to 0.5 A g$^{-1}$, a specific capacity of 163 mA h g$^{-1}$ is restored, thus demonstrating the electrochemical reversibility of the Zn|| $V_d$–$V_2O_3$ cell. Furthermore, we compared the capacity retention rate of different ZIBs electrodes when the specific current was increased tenfold as shown in Supplementary Fig. 14 and Supplementary Table 4. When the specific current increases ten times, the capacity retention rate is still 70.4%, which is better than that of many ZIBs electrodes[33–37], exhibiting good rate capability. We also report that the Zn||$V_d$–$V_2O_3$ cell delivers long cycling stability with a capacity retention rate of 98% after 10,000 cycles, 90% after 20,000 cycles, and 81% after 30,000 cycles at a specific current of 5 A g$^{-1}$ (Fig. 3c). As shown in Supplementary Fig. 15, the life span of $V_d$–$V_2O_3$ is preferable to most of recently reported aqueous ZIBs (Supplementary Table 5)[38–43]. Besides, the Ragone plot showing the specific energy and power compared with other ZIBs cathodes is disclosed in Supplementary Fig. 16. The results show that at a specific power of 332.7 W kg$^{-1}$, the specific energy of the Zn||$V_d$–$V_2O_3$ cell is 110.9 Wh kg$^{-1}$, which is better than many reported ZIBs cathodes, such as Na$_3$V$_2$(PO$_4$)$_3$[44], Na$_{0.95}$MnO$_2$[45], FeFe(CN)$_6$[46], CuHCF[47], ZnHCF[12], VS$_2$[48]. These results certainly highlight the good potentials of $V_d$–$V_2O_3$ cathode-based Zn batteries in the field of electrochemical energy storage devices. As shown in Supplementary Figs. 17 and 18, the 0.5 h-$V_d$–$V_2O_3$ (3.0% vanadium vacancies), 6 h-$V_d$–$V_2O_3$ (4.7% vanadium vacancies), and c-$V_2O_3$ (no vanadium vacancies) cathodes demonstrate inadequate rate and stability electrochemical performance for aqueous ZIBs, which strongly confirms the positive effects of vanadium vacancies in $V_d$–$V_2O_3$ cathode and 5.7% vanadium vacancies has the most appealing electrochemical performance, especially cycle stability (Supplementary Table 6). The long cycle life of this $V_d$–$V_2O_3$ cathode comes ultimately from abundant vacancy clusters that attenuate the strong electrostatic interaction between Zn$^{2+}$ and the $V_d$–$V_2O_3$ host.

To further understand the Zn$^{2+}$ storage performance, the electrochemical kinetics was investigated. As shown in Fig. 3d, CV measurements are carried out at different scan rates. With the increase of scan rates from 0.1 to 1.0 mV s$^{-1}$, the CV curves show a similar shape, and the reduction and oxidation peaks are well preserved. Regularly, the peak currents ($i$) and their corresponding sweep rates ($v$) obey a power-law relationship that is described by

$$i = av^b, \tag{1}$$

where $i$ represents the peak current (A), $v$ represents the scan rate (V s$^{-1}$), and $a$, $b$ are constants. The $b$ values are used as the base for analyzing electrochemical processes. As known, a $b$ value of 0.5 indicates an electrochemical process that governed by ionic diffusion, while a $b$ value of 1.0 indicates a capacitive storage process. From the equation of log ($i$) = $b$log ($v$) + log ($a$) derived from Eq. (1), the calculated $b$ values for both cathode and anode peaks from CV curves are 0.94, 0.75, 0.79, and 0.88, respectively (Fig. 3e). It is suggested that the Zn$^{2+}$ storage behavior in $V_d$–$V_2O_3$ is controlled collectively by ionic diffusion and capacitive processes, which leads to fast Zn$^{2+}$ diffusion kinetics enabling the high-rate performance. In order to further quantify the contribution of diffusion-controlled and capacitive-controlled at a specific scan rate, Eq. (1) is divided into two halves to form formula (2):

$$i(V) = k_1\nu + k_2\nu^{1/2} \tag{2}$$

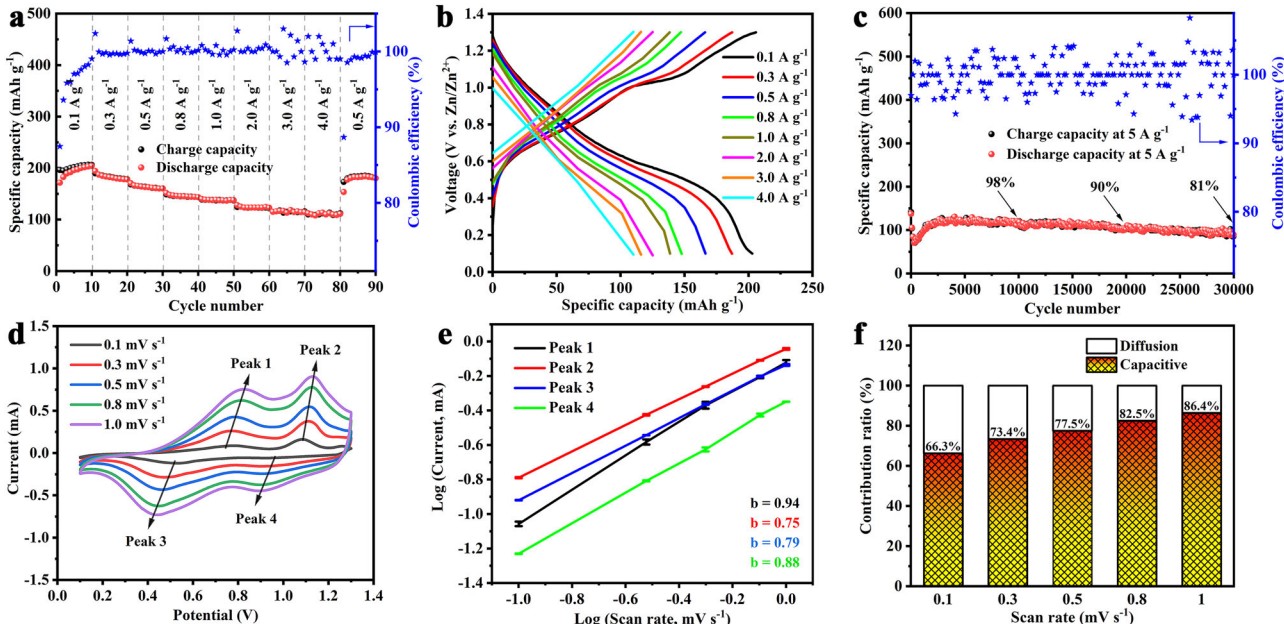

**Fig. 3 Electrochemical energy storage performance of Zn||V$_d$–V$_2$O$_3$ cells. a** Rate performance at different specific currents for V$_d$–V$_2$O$_3$. **b** Galvanostatic discharge profiles of V$_d$–V$_2$O$_3$ at increasing specific currents from 0.1 to 4 A g$^{-1}$. **c** Long-term cycling performance and coulombic efficiency for V$_d$–V$_2$O$_3$ at a specific current of 5 A g$^{-1}$. **d** CV curves of the V$_d$–V$_2$O$_3$ at scan rates ranging from 0.1 to 1 mV s$^{-1}$. **e** Error bars plot of Log (*i*) versus log (*v*) plots at specific peak currents (The $R^2$ values corresponding to the lines of peaks 1, 2, 3, and 4 are 0.9975, 0.9998, 0.9998, and 0.9995, respectively). **f** Contribution ratios of capacitive and diffusion-controlled capacities at different scan rates from 0.1 mV s$^{-1}$ to 1.0 mV s$^{-1}$.

According to the above equation, the current (*i*) at a specific potential (*V*) can be divided into a capacitance limiting effect ($k_1 v$) and a diffusion control effect ($k_2 v^{1/2}$). As shown in Supplementary Fig. 19, the capacitance contribution (corresponding to the purple region) is 82.5% of the overall contribution at a scan rate of 0.8 mV s$^{-1}$. With the increases of scan rates from 0.1 to 1 mV s$^{-1}$, the capacitance contribution rates increase from 66.3% to 86.4% (Fig. 3f). This shows that the proportion of capacitance dominated process is increased, which directly contributes to good rate performance due to the fast kinetics of Zn$^{2+}$. Galvanostatic intermittent titration technique (GITT) is performed to analyze the diffusion coefficient of Zn$^{2+}$ in the Zn|| V$_d$–V$_2$O$_3$ cell (Supplementary Figs. 20 and 21). The result shows that the diffusion coefficient of Zn$^{2+}$ in the V$_d$–V$_2$O$_3$ is between $10^{-7}$ and $10^{-8}$ cm$^2$ s$^{-1}$, which stays ahead of the other existing cathodes[6,49,50].

**Zinc-ion storage mechanism of V$_d$–V$_2$O$_3$.** Density functional theory (DFT) calculations were conducted to explore the function of V$_d$–V$_2$O$_3$ for Zn$^{2+}$ storage. To investigate the distribution of vanadium vacancies, diverse vanadium-defective models at the concentration of 6.25%, in good agreement with the XRD and NPD Rietveld refinement results were constructed as shown in Supplementary Fig. 22. A lower formation enthalpy represents a more stable phase. Structure 1 possesses the smallest formation enthalpy, illustrating the short aggregation of V$_d$ clusters. Then, the pristine V$_2$O$_3$ (p-V$_2$O$_3$) and V$_d$–V$_2$O$_3$ were implemented to disclose the insertion of Zn$^{2+}$. The blocked insertion of Zn$^{2+}$ into p-V$_2$O$_3$ was observed due to the positive Gibbs free energy (2.69 eV)[27], demonstrating almost no capacity contribution (Fig. 4a), which is consistent with the poor Zn$^{2+}$ storage performance of c-V$_2$O$_3$ (Supplementary Fig. 18). Furthermore, the direct insertion of Zn$^{2+}$ into p-V$_2$O$_3$ may cause structural instability. However, for V$_d$–V$_2$O$_3$, vanadium vacancies accept the insertion of Zn$^{2+}$ and provide high capacity than c-V$_2$O$_3$.

Intriguingly, the distinguishing Gibbs free energies demonstrate the process of insertion of Zn$^{2+}$ into V$_d$–V$_2$O$_3$ is different. Firstly, the vanadium defect is occupied with Zn$^{2+}$ and a large amount of heat was released ($-1.34$ eV) due to the strong electrostatic interaction, improving the integrities and stabilities of V$_d$–V$_2$O$_3$. Nevertheless, due to this strong electrostatic interaction, the extraction of this kind of Zn$^{2+}$ is unbearable, demonstrating the self-anchoring action of Zn$^{2+}$ in the lattice. Secondly, the feasible and sustainable insertion of Zn$^{2+}$ into V$_d$–V$_2$O$_3$ is observed, affording the capacity and voltage (Fig. 4a). This phenomenon may lead to the residual of Zn in V$_d$–V$_2$O$_3$. Consequently, the dual-effect of vanadium vacancies in V$_d$–V$_2$O$_3$ is specified in Fig. 4b. When Zn$^{2+}$ initially enters into the V$_d$–V$_2$O$_3$ that has many vanadium vacancies, part of Zn$^{2+}$ will be riveted on vanadium vacancies and caged inside during the whole time. It is worth noting that the coulombic efficiency of the first cycle and the second cycle was compared, which increased from 87.1% to 93.6%, indicating that the V$_d$–V$_2$O$_3$ indeed undergoes a self-optimization process after the first discharge. In other words, the eventual structure is a Zn doped V$_d$–V$_2$O$_3$ after the first discharging self-optimized process, in which the Zn$^{2+}$ can reversibly insert or leave in the subsequent cycles.

Based on the above analysis, a series of characterizations were conducted to demonstrate the Zn$^{2+}$ storage mechanism of the V$_d$–V$_2$O$_3$ cathode and the effect of vanadium vacancies. It can be seen from Fig. 5a, b that the characteristic (104) and (110) peaks move to a lower 2θ degree during the discharging process and return to the original position in the subsequent charge process. These reversible movements originate from the expansion and contraction of the lattice of V$_d$–V$_2$O$_3$ with the de/intercalation of Zn$^{2+}$. Besides, no other diffraction peaks were detected, indicating no phase transformations in the V$_d$–V$_2$O$_3$ electrodes during the charge/discharge process. The stability of V$_d$–V$_2$O$_3$ was further verified via XRD measurements after 500 cycles (Supplementary Fig. 23). It is noteworthy that the morphology of the V$_d$–V$_2$O$_3$ electrodes have transformed into particles with a

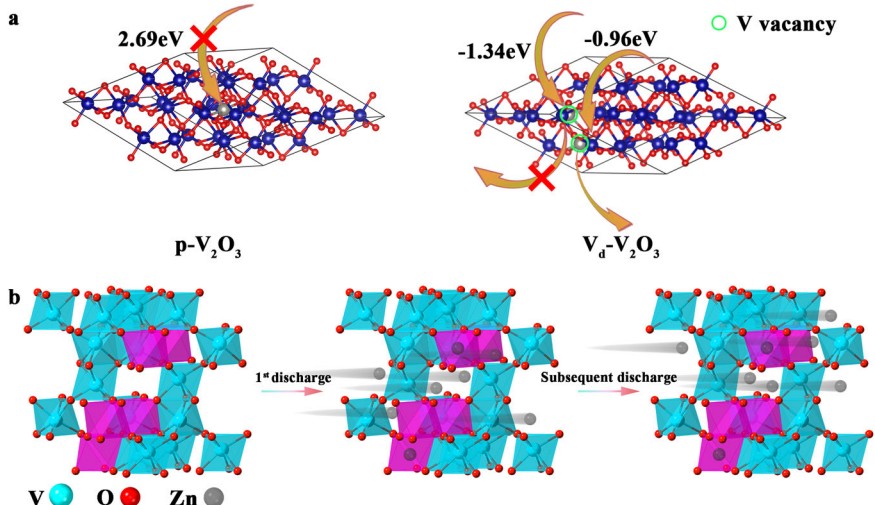

**Fig. 4 Zn-ion storage mechanistic investigation in Zn||V$_d$-V$_2$O$_3$ cells. a** The Gibbs free energy of different models. **b** The schematic illustration of the energy storage mechanism in the **Zn||V$_d$–V$_2$O$_3$ cells**.

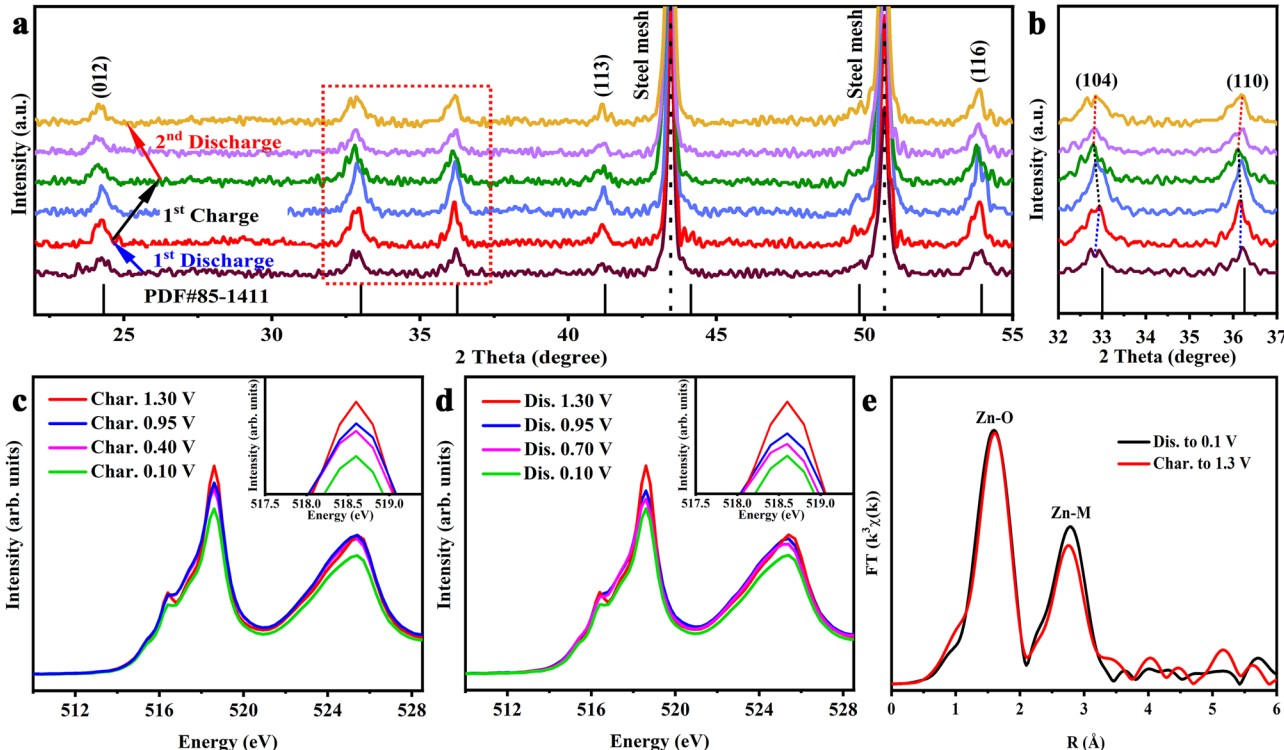

**Fig. 5 Ex situ V$_d$–V$_2$O$_3$ electrode measurements. a** Ex situ XRD patterns of V$_d$-V$_2$O$_3$ electrodes at different cut-off voltages during the charge and discharge process. **b** An enlarged view of the red dotted frame in Fig. 4a. **c** Normalized XANES spectrum of the V L-edge for V$_d$-V$_2$O$_3$ electrodes at different voltages during the charging process. **d** Normalized XANES spectrum of the V-edge for V$_d$-V$_2$O$_3$ electrodes at different voltages during the discharging process. **e** Fourier-transformed Zn K-edge EXAFS spectra of V$_d$-V$_2$O$_3$ electrodes at fully charged and discharged states.

diameter of ~25 nm after the first cycle and remained unchanged in the subsequent cycles, which adapts to the reversible insertion/ extraction of Zn$^{2+}$ better. (Supplementary Figs. 24–26).

XPS and soft X-ray absorption spectroscopy (sXAS) measurements were performed to give insight into the chemical states of the V$_d$–V$_2$O$_3$ electrodes at different states. The reversible insertion and extraction of Zn$^{2+}$ are shown in Supplementary Fig. 27. At a fully discharged state, the V$_d$–V$_2$O$_3$ electrode displays two Zn 2$p_{3/2}$ components located at 1022.5 eV and 1023.2 eV which belong to the intercalated Zn$^{2+}$ at different

occupation sites (vanadium vacancies and tunnels nearby vanadium vacancies). At a fully charged state, the Zn 2$p_{3/2}$ peak located at 1023.2 eV disappears while the peak of 1022.5 eV is preserved. That is to say that some vanadium vacancies occupying Zn$^{2+}$ are riveted in the lattice of V$_d$–V$_2$O$_3$, only enabling Zn$^{2+}$ reversibly (de)intercalation in the tunnel neighboring the remaining vanadium vacancies. Precise quantification of the V and Zn contents at fully dis/charged states was carried out via inductively coupled plasma emission spectroscopy (ICP-AES) measurements and the detailed results shown in

Supplementary Table 7. The results are verified via TEM elemental mapping where the Zn, V, and O elements are uniformly distributed (Supplementary Figs. 28 and 29). The charge compensation of V in the process of $Zn^{2+}$ insertion/extraction is shown in the V L-edge patterns (Fig. 5c, d). Two peaks located at ~518 eV and 525 eV are observed in the V L-edge pattern, corresponding to V $2p_{3/2} \rightarrow$ V $3d$ and V $2p_{1/2} \rightarrow$ V $3d$ transitions, respectively[51,52]. The intensity of V $2p_{3/2}$ peaks are increased with the entrance of $Zn^{2+}$ and decreased with the release of $Zn^{2+}$, gradually, stating the electronic acquirement and deprivation of V in the $Zn^{2+}$ electrochemistry. Notably, under the same voltage, the peak intensity remains the same revealing the high reversibility of the $V_d$–$V_2O_3$ electrode. The valence changes in the whole cycle are also shown in the high-resolution XPS of V (Supplementary Fig. 30) where the peak of $V^{4+}$ becomes dominant upon charging, while releasing a sign of a let-up upon discharging. Given the local environment of vacancy, the local chemical and electronic environment of intercalated $Zn^{2+}$ was investigated by Zn K-edge XAFS. Since the surrounding local environment of $Zn^{2+}$ is the same during charging and discharging, the K-edge XAFS of Zn is changeless (Fig. 5e and Supplementary Fig. 31).

In summary, we have quantified 5.7% $V_d$ clusters in a $V_d$–$V_2O_3$ cathode for aqueous ZIBs which shows appealing $Zn^{2+}$ storage performance. The DFT calculations indicated that the $Zn^{2+}$ storage reversibility and stability improved under the effects of $V_d$ clusters. In detail, part of vanadium vacancies provides permanent sites for the preoccupation of a small amount of $Zn^{2+}$ so that the system could pose a more stable structure to avoid deterioration during the process of $Zn^{2+}$ insertion/extraction. Meanwhile, the other vanadium vacancies can effectively weaken the strong interaction between $Zn^{2+}$ and the $V_2O_3$ material host to allow free insertion/extraction of $Zn^{2+}$. Benefitting from the 5.7% $V_d$ clusters, the $V_d$–$V_2O_3$ cathode achieved a capacity of 196 mA h $g^{-1}$ at 0.1 A $g^{-1}$, and exhibited long stability up to 30,000 cycles with a capacity retention of 81%. This accurately quantifying and determining the effect of defects opens a window for designing aqueous ZIBs cathodes with long stability.

## Methods

**Preparation of $V_d$–$V_2O_3$.** The $V_d$–$V_2O_3$ was prepared by a hydrothermal method. First of all, 0.1856 g of vanadyl acetylacetonate and 0.42 g of urea were dissolved in 30 ml of ethylene glycol. Subsequently, 5 ml $H_2O_2$ (6 wt%) was poured into the above-mentioned solution and stirred evenly. After that, the obtained solution was transferred to a Teflon-lined sealed autoclave (50 ml) and kept at 200 °C for 8 h. After natural cooling, the obtained precursor was washed five times with deionized water and ethanol, and then freeze-dried for further use. The freeze-dried powder was calcined at 600 °C for 2 h with a heating rate of 5 min °C$^{-1}$ in argon mixed with 20% hydrogen atmosphere. After the above process, black $V_d$–$V_2O_3$ powder was obtained.

**Materials characterizations.** SEM images and TEM images were observed by a cold field SEM (SU8220) and a transmission electron microscopy (TEM: JEOL JEM2010), respectively. The chemical states were detected by EPR on a tyFA200. XPS results were obtained from electron energy disperse spectroscopy (ESCALAB 250) with monochrome Al anode (Al Kα = 1486.6 eV). The binding energy had been corrected by carbon energy of 284.6 eV. The nitrogen adsorption/desorption isotherms were obtained on a TriStar II 3020 analyzer. The TGA was performed on Q5000 at 10 °C min$^{-1}$ under air atmosphere. Plasma Atomic Emission Spectrometer (Optima 7300 DV) was used to determine the content of elements. The V L-edge was measured at the beamline BL07W of the Hefei Synchrotron Radiation Equipment. The V K-edge was obtained on the beamline 1W1B in Beijing Synchrotron Radiation Facility (BSRF). The X-ray was monochromatized by a double-crystal Si (111) monochromator, and the energy was calibrated using a V metal foil for V K-edge. XAFS data were analyzed with the WinXAS3.1 program[53]. The Zn K-edge was got from the beamline 14W1 of the Shanghai Synchrotron Radiation Facility (SSRF). The X-ray was monochromatized by a double-crystal Si (111) monochromator, and the energy was calibrated using a Zn metal foil for Zn K-edge. XAFS data were analyzed with the WinXAS3.1 program[53]. Theoretical amplitudes and phase-shift functions of Zn–O and Zn–Zn were calculated with the

FEFF8.2 code[54] using the crystal structural parameters of the ZnO. XRD patterns were obtained on a Sample horizontal high-power X-ray diffractometer (Rigaku TTRIII) with Cu radiation ($\lambda = 1.54$ Å) in a range of 5–120°. NPD data were collected at the beamline of general purpose powder diffractometer (GPPD) at the China Spallation Neutron Source (CSNS). The XRD and NPD data were refined by the Rietveld method using the General Structure Analysis System[55] (GSAS) suite of programs with the EXPGUI[56] interface. The PALS measurements were carried out at SKLPDE (State Key Laboratory of Particle Detection and Electronics) of USTC. The PAL spectra were collected using a fast-fast coincidence system with a time resolution of around 200 ps in full width at half maximum (FWHM). The prepared sample powder was pressed into disc-shaped pellets with a diameter of 8 mm and a thickness of ~2 mm. A $^{22}$Na positron source (~30 μCi, sealed between two Kapton films of 0.0075 × 10 × 10 mm) was sandwiched between two identical pellet-shaped samples. The sample-source-sample set was fixed in a chamber that was evacuated by a turbo molecular pump. The total channel number is 4096, and the channel width is 10.417 ps/channel. The total counts of 2 × 10$^6$ were collected for each PAL spectrum. The PAL spectra were decomposed using the LTv9 software[57].

**Measurements of electrochemical performance.** All the electrodes used in this experiment were made by the following steps: (1) The active material of $V_d$–$V_2O_3$ (~40 mg), Ketjen Black and polyvinylidene fluoride (PVDF) were weighed at the mass ratio of 7:2:1, and then mixed and ground in an agate mortar for 15 min. (2) 250 μL N-methyl-2-pyrrolidone (NMP) was added into the agate mortar with further grinding for 5 min. (3) After grinding completely and uniformly, the resultant slurry was coated on the pretreated stainless steel mesh (The stainless steel mesh was pretreated as follow: first, the stainless steel mesh of 1000-mesh was punched into circles with a radius of 6 mm, then immersed in ethanol solution to ultrasonic wash for 5–10 min, and finally put it in a 100 °C vacuum to dry for 12 h). The mass loading of active material on each stainless steel mesh is ~1.13 mg cm$^{-2}$. (4) The coated stainless steel mesh was put into a 70 °C oven for pre-drying for 30 min and then moved into a 100 °C vacuum oven to dry for 12 h. The positive shell, the above-mentioned dried electrode, type 102 medium speed qualitative filter paper (separator) with a radius of 10 mm, 120 μL 3 M Zn(CF$_3$SO$_3$)$_2$ (98% pure electrolyte salt), zinc sheet (counter electrode), gasket, shrapnel, and negative shell were assembled into a CR-2032 coin-cell in above order. The metal purity of zinc sheet purchased from Chengshuo Company is not <99.99%, whose radius and thickness are respectively 6 mm and 30 μm. The rate performance, stability performance, and GITT of the cells were tested in an environmental chamber under the Land CT2001A battery test system in the voltage range of 0.1–1.3 V at 25 °C. The GITT curves were charged and discharged for 1 min at a specific current of 150 mA g$^{-1}$, then relaxed for 30 min to reach the quasi-equilibrium potential. The CV curves at different scan rates were performed on the electrochemical workstation (CHI660D, Shanghai CH Instrument Company, China) in the voltage range of 0.1–1.3 V. Before performing the ex-situ electrode measurements, the assembled cells after cycled at different cut-off voltages were disassembled in an environmental chamber at 25 °C. The electrodes taken out of the cells were rinsed with ethanol 4–5 times. Afterward, the cleaned electrodes were dried in an 80 °C vacuum drying oven, and then transferred to a glove box filled with argon, and taken it out before the measurements.

**DFT calculation.** First-principles calculations were performed by using DFT[58] based on project augmented wave method (PAW)[59] which is implemented in the Vienna ab initio Simulation Package (VASP)[60]. In order to circumvent the over delocalization of the 3d-electrons in metal oxides, the DFT + U ($U = 3.25$ eV) method is implemented[61]. Perdue-Burke-Ernzerhof (PBE)[62] version of the generalized gradient approximation (GGA) is performed to deal with the exchange and correlation energy. The Brillouin zone is sampled by using 3 × 3 × 1 based on Monkhorst-Pack k-point mesh. A plane-wave basis set with a cutoff energy of 450 eV is efficient to guarantee the convergence of the total energy. The atoms in the structure are completely relaxed with force below 0.01 eV Å$^{-1}$.

## Data availability

All the relevant data are included in the paper and its Supplementary Information. Source data are provided with this paper.

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

## Acknowledgements

This work was financially supported in part by the National Key R&D Program of China (2020YFA0405800-L.S.), NSFC (U1932201-L.S., U2032113-S.M.C., and 22075264-S.M.C.), CAS Collaborative Innovation Program of Hefei Science Center (2019HSC-CIP002-S.L., 2020HSC-CIP002-S.M.C.), CAS International Partnership Program

(211134KYSB20190063-S.L.), and USTC Research Funds of the Double First-Class Initiative (Grant No. YD2310002003-S.M.C.). L.S. acknowledges the support from Institute of Energy, Hefei Comprehensive National Science Center, University Synergy Innovation Program of Anhui Province (GXXT-2020-002). The authors thank the Beijing Synchrotron Radiation Facility (1W1B, 4W1B and 4B9A, BSRF), Shanghai Synchrotron Radiation Facility (BL14W1 and 14B1, SSRF), the Hefei Synchrotron Radiation Facility (MCD-A and MCD-B Soochow Beamline for Energy Materials, Infrared spectroscopy and microspectroscopy at NSRL), and the USTC Center for Micro and Nanoscale Research and Fabrication for helps in characterizations.

## Author contributions

L.S. and S.M.C. supervised the project. K.F.Z. and S.Q.W. designed the work and carried out most of the experiments. L.H.H. and F.R.S. measured and analyzed XRD and NPD data. H.W.S. simulated most of the calculations. P.J.Z. and C.D.W. performed XAFS and sXAS experiments. Y.Y.C. and X.G. helped to prepare most of the samples. H.J.Z., B.J.Y., and M.L. guided the PALS measurements and helped to analyze PALS results. X.J.W. helped to explain some experimental data. All the authors discussed the results and assisted during the manuscript preparation.

## Competing interests

The authors declare no competing interests.
