## [Peer Review File · Nature Communications]

REVIEWER COMMENTS

Reviewer #1 (Remarks to the Author):

The authors report a combined method for quantifying vanadium-vacancy in V₂O₃, which is beneficial for ultra-long cycling aqueous Zn-ion battery (ZIB). They show very interesting results, but more systematic experiments are needed to convince their claims.

1. Is vanadium-vacancy in V₂O₃ only good for improving the cycle stability of ZIB? Vd-V₂O₃ shows a much lower capacity than other V₂O₃-based or vanadium oxide-based electrodes. The authors compared only the cycle stability.
2. The authors need to include the following reference that shows much better electrochemical performances of the V₂O₃ electrode. The reference shows a much higher capacity and rate capability than the manuscript and offers similar cycle stability. "Anodic Oxidation Strategy toward Structure-Optimized V₂O₃ Cathode via Electrolyte Regulation for Zn-Ion Storage", ACS Nano 2020, 14, 6, 7328–7337. The authors should update the references to the latest.
3. How was the vanadium-vacancy created? Can the degree (concentration) of vanadium-vacancy in V₂O₃ be controlled?
4. Is the proposed method reliable to quantify the vanadium-vacancy in V₂O₃? The authors just assumed that c-V₂O₃ had no vanadium-vacancy. Can the authors prove that there is no vanadium-vacancy in c-V₂O₃ using the proposed method?
5. Electrochemical performance other than cycle stability of Vd-V₂O₃ should also be compared with others.
6. The diffusion coefficient of Zn²⁺ in the Vd-V₂O₃ electrode is 10⁻⁷ to 10⁻⁸ cm²/s, which is faster than other electrodes, including the above reference, but the rate capability of the Vd-V₂O₃ electrode is lower than that of the other electrodes. What would be the reason? Due to the low electrical conductivity of the Vd-V₂O₃ electrode?

Reviewer #2 (Remarks to the Author):

Manuscript ID: NCOMMS-21-15134

Manuscript Title: Quantifying vanadium-vacancy clusters in V₂O₃ towards ultra-long cycling aqueous zinc-ion battery

The paper deals with an important topic related to improving the performance of rechargeable aqueous Zinc ion batteries through the structural defects in cathode materials. The authors investigated vanadium defect (Vd) clusters in the V₂O₃ lattice by combining spectroscopy analytical techniques (Neutron powder diffraction (NPD), X-ray powder diffraction (XRD), positron annihilation spectroscopy (PAS) and synchrotron-based X-ray) and theoretical calculations (Rietveld refinement and density functional theory (DFT) calculation). The authors demonstrated that Vd clusters can provide permanent sites for Zn²⁺ anchormen to enhance the integrity of V₂O₃ after the first discharging process. It also make Zn²⁺ de/intercalation in complex oxide, which contributes in reducing the strong electrostatic interaction between host multivalent ions. They authors correlate the Vd defect clusters with the with ultra-long cycle life of Vd-V₂O₃ cathode for Zn²⁺ storage. A full complete set of experimental data and theoretical calculations was provided with good interpretation of these results.

The reviewer finds that the novelty of the work in quantifying the structural defects in cathode material and the enegineering design of the Vd-V₂O₃ cathode material is unique in this work. The paper can be accepted for publication in nature communications after considering the following changes:

- 1- Abstract.

- The following sentence "Besides, the density functional theory (DFT) calculations storage performance of Zn²⁺" is very long. Please split in 2-3 sentences for clarification.

2- Introduction.

- "... security issues...": Please change the word "security" with more appropriate word ("safety" or "hazard").

- "... ZIBs have great commercial potential. But great upgrading ...": The word "great" is used consecutively twice. Please replace one of the words "great" with another word having the same meaning (important, huge, suitable, ...).

- "Accurately quantification ...": Change "Accurately" with "Accurate"

- "For example, light oxygen is difficult to be detected by XRD owing to its small atomic radius, compared with most metal elements.": Please support this idea with references (example: Chem. Sci.,2021,12,517–532).

- "For example, light oxygen is difficult to be detected by XRD owing to its small atomic radius, compared with most metal elements. Consequently, the joint application of multiple spectroscopy is essential to accurately quantify defects.": The transition between ideas is not evident. Please revise part to clearly state the novelty of your work.

3. Structure and morphology characterization of Vd-V₂O₃

- "But the situation in Vd-V₂O₃ is different, the surface of Vd-V₂O₃ is proved to be uniformly coated with carbon" : The reviewer did not notice a difference in carbon content in both Vd-V₂O₃ c-V₂O₃ materials? For c-V₂O₃ material, the carbon content should be provided from the supplier if it exists. The TGA analysis of c-V₂O₃ material did not show that carbon exist in its composition. Please clarify this result. Can the carbon content be controlled during hydrothermal treatment and annealing? Do the author use any other source of vanadium for comparison? If yes, please provide the data of mapping analysis.

- Are the provided data in Figure Supplementary Figure 9 are the best fitting for the Fourier transformed EXAFS? The authors suggested that CN less than 3 for V in Vd-V₂O₃ to V vacancies. Please support your suggestion with references from literature reported for V or any other transition metal oxides.

- PAS analysis results can be more detailed. Vd-V₂O₃ and c-V₂O₃ showed similar I₃ intensity. However, differences can be observed in I₁ and I₂ intensities between Vd-V₂O₃ and c-V₂O₃. The authors focused only on I₂. Please provide a full interpretation of the PAS results.

- Consistent abbreviation is required: Use PAS or PALS?

- Please provide NPD and XRD spectra of c-V₂O₃ (as supplementary material) to observe the differences with Vd-V₂O₃.

- "In general, we have demonstrated the presence of coordinately unsaturated atoms of V in Vd-V₂O₃ and the vacancy occupancy rate of vanadium is 5.7%, and the vanadium vacancies exist in the form of vacancy clusters.": Please revise this sentence to make a reasonable conclusion.

4. Electrochemistry

- The authors used both ZnSO₄ and Zn(CF₃SO₃)₂ supporting electrolytes in their electrochemical tests. One sentence should be added to explain the differences between them and the effect of the concentration of the supporting electrolyte.

- The specific capacity of Vd-V₂O₃ is lower than other cathode materials reported in the literature. This should be clearly stated.

- "The longevity of this Vd-V₂O₃ cathode comes ultimately from abundant vacancy clusters that attenuate the strong electrostatic interaction between Zn²⁺ and the Vd-V₂O₃ host.": This conclusion is not well placed in the manuscript. It should be moved to the end.

- "the c-V₂O₃ cathode, without vacancies, ...": This is wrong based on PAS results. Please revise the sentence for accuracy.

5- Zinc-ion storage mechanism of Vd-V₂O₃

- Can the authors provide a clear explanation of the difference in V defects between the DFT model (6.25%) and experimental data (5.7%)?

- Please number the structures in Supplementary Figure 18.

- The abbreviation p-V₂O₃ should be provided the first time is cited in the text.

- "a large number of heat ...": Please change "number" with "amount" or "quantity"

- "When Zn²⁺ initially enters ...": Please replace "entries" with "enters"
- "the eventual structure is a Zn doped Vd-V₂O₃ after the first discharging self-optimized process, in which the Zn²⁺ will reversibly insert or leave in the subsequent cycles.": Please correlate this suggestion with the initial coulombic efficiency of Zn/Vd-V₂O₃ cell.

Reviewer #3 (Remarks to the Author):

Comments:

I have reviewed the article titled "Quantifying vanadium-vacancy clusters in V₂O₃ towards ultra-long cycling aqueous zinc-ion battery". The authors have carried out exhaustive spectroscopy and diffraction studies in order to establish the presence of vanadium cluster vacancies in the V₂O₃ cathode material. Although many studies were done, there is no categorical proof that the presence of vacancy is what gave rise to capacity and good cycle life for the zinc-ion battery. The authors fail to establish a strong correlation between vacancy and battery performance, they should have made a series of V₂O₃ having varying degree of vacancy and correlated its effect to the battery performance. Since a number of studies have been done in this work, it should have been submitted as a full manuscript instead of communication. In order to make it look like a communication, authors have curtailed the discussion part significantly without providing crucial information. Moreover, the performance of the battery is not very impressive, I could find even organic cathode materials with similar capacity and able to operate upto 63C rate (ACS Appl. Energy Mater. 2021, 4, 1218–1227). My specific comments are given below.

1. Replace "GTA" with "TGA" in the caption of supplementary Fig. 8
2. TGA of c-V₂O₃ shows an increase in wt. % upto 19.8%, whereas it is stated otherwise in the text. Why would the wt.% increase with the increase in temperature? It is not clear whether it is performed in an oxygen environment or something else.
3. It is stated by authors that "carbon on vd-V₂O₃ is not easily oxidized", but the data shows a drop in wt.% with temperature, hence the statement is not matching with the figure.
4. Fitting in Fig. 9 is not satisfactory beyond 3 A, is there any specific reason?
5. It is indicated the presence of two pairs of redox peaks located at 1.09/0.78 V and 0.93/0.53V, which is not matching with supplementary Fig. 13 and redox pairs are not clear. I see redox pairs at 0.7/0.55 and 1.1/0.9 V.
6. In supplementary fig 13, the area under the first cycle is lower than the later two cycles. How reproducible is this data? Why would the area increase with cycle? Do authors repeat this experiment and confirm the results?
7. Does vd-V₂O₃ undergoes any surface change due to cycling? Any in-situ evidence?
8. Authors claim two-step (de)intercalation of Zn²⁺, what are those two sites that promote (de)intercalation at two different potentials.?
9. In Fig. 2, it is not clear, what the blue dots and red dots correspond to? Which one is for coulombic efficiency and which one represents capacity?
10. The maximum specific capacity obtained is ~ 187 mAhg⁻¹ at 0.1 A/g. The rate and capacity values are bit low, comparable to the many reports in the literature.
11. According to the authors, the calculated b values for both cathode and anode peaks from CV curves are 0.94, 0.75, 0.79 and 0.88, respectively. This suggests the Faradaic component of charging and discharging are not equal, then how could authors cycle without issues for long.?
12. With the increase in scan rate, the capacitive contribution increasing (Fig. 2f), if so what is the use of having Vd-vacancy in the bulk and especially for high rate performance.?
13. In supplementary Fig. 17, why the D_{Zn²⁺} values do not follow a trend? Give complete details of the calculation involved in estimating D_{Zn²⁺}. (parameters and their values).

Responses to Reviewer's Comments

Dear Reviewers,

We would like to thank you for your critical assessment of our submission. We have carefully considered all comments received and have modified our manuscript correspondingly. Additional data and revised plots are provided to clarify and resolve the issues pointed out in the original manuscript, as well as the overall restructuring of the discussion. With these revisions, we hope that we have satisfactorily addressed all concerns and that the current version is now acceptable for publication in **Nature Communications** journal.

Below please find our point-by-point response to the comments received. The major changes in our revised manuscript have been marked as **Red**.

Reviewer #1 (Remarks to the Author):

The authors report a combined method for quantifying vanadium-vacancy in V_2O_3 , which is beneficial for ultra-long cycling aqueous Zn-ion battery (ZIB). They show very interesting results, but more systematic experiments are needed to convince their claims.

We highly appreciate the reviewer for the useful suggestions. Accordingly, we have added new data and corrected descriptions, and carefully modified our manuscript. We would like to reply to the reviewer's concerns one by one as below.

1. Is vanadium-vacancy in V_2O_3 only good for improving the cycle stability of ZIB? V_d - V_2O_3 shows a much lower capacity than other V_2O_3 -based or vanadium oxide-based electrodes. The authors compared only the cycle stability.

Reply: Thanks for pointing out this issue.

(1) According to the experimental and DFT calculations, we find that the contribution of vanadium vacancies to the capacity and rate performance of the V_d - V_2O_3 electrode is restricted, but can effectively promote the cycle stability of ZIB, and make it exhibit ultra-long cycle stability. The expound is as follows.

Through X-ray powder diffraction (XRD), soft X-ray absorption spectrum (sXAS), and X-ray photoelectron spectroscopy (XPS), we have certified that the structure and valence changes of the V_d - V_2O_3 electrode are extremely reversible during the charge and discharge process, which reflects the stability of V_d - V_2O_3 . Besides, the DFT calculations have further indicated that the Zn^{2+} storage reversibility and stability will be greatly enhanced under the effects of V_d clusters. Concretely, part of vanadium vacancies provides permanent sites for the preoccupation of a small amount of Zn^{2+} so that the system will have a more stable structure to against collapsing during the process of Zn^{2+} insertion/extraction. Meanwhile, the other vanadium vacancies can effectively weaken the strong interaction between Zn^{2+} and the V_2O_3 material host to allow free insertion/extraction of Zn^{2+} . Benefitting from the 5.7% V_d clusters, the V_d - V_2O_3 cathode exhibits ultra-long stability up to 30,000 cycles with a capacity retention of 81%, which is the longest aqueous ZIBs stability as far as we know.

To study the effect of the vacancy on capacity and rate performance, V_2O_3 with different vacancy concentrations was synthesized by controlling the calcination time. Under the same other conditions, we altered the calcination time for 0.5 h and 6 h (0.5 h- V_d - V_2O_3 and 6 h- V_d - V_2O_3) respectively. The structure information of 0.5 h- V_d - V_2O_3 and 6 h- V_d - V_2O_3 was also revealed by Rietveld refinement with combined XRD and Neutron powder diffraction (NPD) patterns. The refined results are shown in **Supplementary Table 3**, which exhibits that the structures of 0.5 h- V_d - V_2O_3 and 6 h- V_d - V_2O_3 are the same as the V_d - V_2O_3 . Nevertheless, the occupation ratios of vanadium atoms in 0.5 h- V_d - V_2O_3 and 6 h- V_d - V_2O_3 are 97.0(4)% (3.0% vanadium vacancies) and 95.3(1)% (4.7% vanadium vacancies), respectively, while no oxygen vacancies are found (**Supplementary Figs. 13 and 14**), indicating that the vanadium vacancies concentration at both calcination temperatures is less than that of V_d - V_2O_3 . The V_2O_3 at these two calcination temperatures was made into ZIB electrodes, both of which have inferior cycle stability but are similar to the capacity and rate performance of V_d - V_2O_3 (**Supplementary Fig. 22**). Consequently, it can be concluded that the contribution of vanadium vacancies to the capacity and rate performance of V_d - V_2O_3 electrode is finite

but can effectively improve the reversibility and stability of Zn²⁺ storage, and make it achieve excellent cycle stability.

Supplementary Table 3 | Structural parameters from Rietveld refinement for V_d-V₂O₃.

Sample	Lattice parameters			Atomic occupancy		wRp	χ^2
V _d -V ₂ O ₃	a=4.94734(9) Å	$\alpha=90^\circ$	V=0.296 74 nm ³	V	O	wRp(NDP) =2.95%	22.5
	b=4.94734(9) Å	$\beta=90^\circ$		94.3(1)%	102.7(3)% $\approx 100\%$	wRp(XRD) =3.6%	
	c=13.9990(5) Å	$\gamma=120^\circ$					
c-V ₂ O ₃	a=4.95747(9) Å	$\alpha=90^\circ$	V=0.298 26 nm ³	V	O	wRp(NDP) =3.76%	5.89
	b=4.95747(9) Å	$\beta=90^\circ$		100.7(6)%	99.9(1)%	wRp(XRD) =13.77%	
	c=14.01323(8) Å	$\gamma=120^\circ$					
0.5 h-V _d -V ₂ O ₃	a=4.95594(7) Å	$\alpha=90^\circ$	V=0.298 20 nm ³	V	O	wRp(NDP) =3.58%	2.86
	b=4.95594(7) Å	$\beta=90^\circ$		97.0(4)%	100.5(1)%	wRp(XRD) =12.55%	
	c=14.0192(4) Å	$\gamma=120^\circ$					
6 h-V _d -V ₂ O ₃	a=4.95720(6) Å	$\alpha=90^\circ$	V=0.298 40 nm ³	V	O	wRp(NDP) =3.71%	2.49
	b=4.95720(6) Å	$\beta=90^\circ$		95.3(1)%	100.0(3)%	wRp(XRD) =10.72%	
	c=14.0214(3) Å	$\gamma=120^\circ$					

Supplementary Figure 13 | Neutron powder diffraction and X-ray diffraction refinement patterns for 0.5 h-V₄-V₂O₃. Observed (black circle), calculated diffraction patterns (red line), their difference (purple line), and peak position (black bar) of the NPD pattern (upper part) and XRD pattern (lower part).

Supplementary Figure 14 | Neutron powder diffraction and X-ray diffraction refinement patterns for 6 h-V₄-V₂O₃. Observed (black circle), calculated diffraction patterns (red line), their difference (purple line), and peak position (black bar) of them NPD pattern (upper part) and XRD pattern (lower part).

Supplementary Figure 22 | a, Rate performance at different current densities of 0.5 h- V_d - V_2O_3 electrode. **b**, Rate performance at different current densities of 6 h- V_d - V_2O_3 electrode. **c**, Cycling performance and coulombic efficiency of 0.5 h- V_d - V_2O_3 electrode at a current density of 5 A g⁻¹. **d**, Cycling performance and coulombic efficiency of 6 h- V_d - V_2O_3 electrode at a current density of 5 A g⁻¹.

(2) The elaboration that the V_d - V_2O_3 shows a much lower capacity than other V_2O_3 -based or vanadium oxide-based electrodes is separated into two parts, namely the V_2O_3 -based electrodes and the vanadium-based oxide electrodes.

① The results for the V_d - V_2O_3 show a much lower capacity than other V_2O_3 -based electrodes mainly attributes to the following three points.

First of all, the voltage window is wider than the voltage window in our manuscript, resulting in higher capacity. For example, the voltage window of the references (*ACS Nano* 2020, 14, 6, 7328-7337; *Small Methods* 2021, 2100578) is measured within the range of 0.2-1.6 V. Accordingly, as a comparison, we expanded the test voltage to 1.6 V and found that charging curves similar to that in the references appeared during the first cycle of charging. After the first cycle of charging, the V_2O_3 capacity increased

sharply from 187 mA h g⁻¹ at 0.1 A g⁻¹ to 422 mA h g⁻¹ (**Fig. R1a**). Furthermore, the CV curves also arose similar oxidation peaks at the first cycle of charging to that in the references (**Fig. R1b**). Nevertheless, the oxidation of V_d-V₂O₃ at 1.6 V voltage will bring about structural changes (**Fig. R1c**), and the vanish of vanadium vacancies, which cannot provide excellent stability. After less than 1800 cycles, the capacity retention rate is only 86.2% (**Fig. R1d**). The reference (*ACS Nano* 2020, 14, 6, 7328-7337) also compared the charging to 1.3 V and 1.6 V respectively, and it is found that the capacity does change greatly. Consequently, we did not expand the voltage window to 1.6 V. Secondly, the electrode material undergoes a phase change during the cycle. For instance, the electrode material is converted from V₂O₃ material to V₂O_{5-x}·nH₂O after the first cycle of anodic oxidation in the reference (*ACS Nano* 2020, 14, 6, 7328-7337), which is actually the capacity of V₂O_{5-x}·nH₂O. Furthermore, the V₂O₃ phase disappears after the initial charge process, and Zn_{3+x}(OH)_{2+3x}V_{2-x}O_{7-3x}·2H₂O and zinc vanadate (Zn_yVO_z) phases arise from the second cycle in the reference (*Small Methods* 2021, 2100578). However, the structure of our V₂O₃ electrode has not changed after many cycles (**Supplementary Fig. 27**). Last but not least, the reference (*ACS Nano* 2020, 14, 6, 7328-7337) stated that carbon materials are unsuitable for the storage of Zn²⁺, and the carbon content was excluded in the calculation of the capacity. Nevertheless, we include the carbon content in the calculation, which will also cause the V_d-V₂O₃ electrode to have a lower capacity than the reference.

② Compared with many other vanadium-based oxide electrodes [*Sci. Adv.*, 2019, 5, eaax4279; *Energy Environ. Sci.*, 2021, 14, 4095-4106; *Energy Environ. Sci.*, 2019, 12, 2273-2285; *Angew. Chem. Int. Ed.*, 2019, 58, 7062-7067; *Adv. Mater.*, 2018, 30, 1703725; *Adv. Mater.*, 2018, 30, 1705580], the voltage window is wider than the V_d-V₂O₃ electrode bringing about higher capacity. Moreover, the voltage of these vanadium-based oxide electrodes is not less than 1.6 V, while the V_d-V₂O₃ electrode will oxidize at 1.6 V, resulting in the disappearance of vanadium vacancies and structural changes, which will affect its ultra-long cycle stability.

Fig. R1 | **a**, Rate performance at different current densities of V_d-V₂O₃ electrode in 0.1-1.6 V voltage window. **b**, CV curves of V₂O₃ in Zn(CF₃SO₃)₂ electrolyte at 0.1 mV s⁻¹. **c**, XRD pattern of the V_d-V₂O₃ electrode charged to 1.6 V after 10 cycles. **d**, Long-term cycling performance and coulombic efficiency of V_d-V₂O₃ electrode at a current density of 5 A g⁻¹ in 0.1-1.6 V voltage window.

Supplementary Figure 27 | XRD patterns of V_d-V₂O₃ electrodes under different cycles.

2. The authors need to include the following reference that shows much better electrochemical performances of the V_2O_3 electrode. The reference shows a much higher capacity and rate capability than the manuscript and offers similar cycle stability. “Anodic Oxidation Strategy toward Structure-Optimized V_2O_3 Cathode via Electrolyte Regulation for Zn-Ion Storage”, ACS Nano 2020, 14, 6, 7328-7337. The authors should update the references to the latest.

Reply: Thanks for pointing out this issue. The latest references have been added. The reasons why the manuscript shows a lower capacity and rate capability than the reference has been discussed above.

3. How was the vanadium-vacancy created? Can the degree (concentration) of vanadium-vacancy in V_2O_3 be controlled?

Reply: Thanks for your useful suggestion.

The vanadium vacancies are created in the calcination process, and equally the concentration of vanadium-vacancy in V_2O_3 can be controlled by calcination time. The detailed elaborations are as follows.

Assuming that the calcination time can control the vacancy concentration, we did a piece of comparative experiments. Under the same other conditions, we controlled the calcination time for 0.5 h and 6 h (0.5 h- V_d - V_2O_3 and 6 h- V_d - V_2O_3) respectively. The structure information of 0.5 h- V_d - V_2O_3 and 6 h- V_d - V_2O_3 was revealed by Rietveld refinement with combined XRD and NPD patterns. The refined results are shown in **Supplementary Table 3** above, which displays that the structures of 0.5 h- V_d - V_2O_3 and 6 h- V_d - V_2O_3 are the same as the V_d - V_2O_3 . Nevertheless, the occupation ratios of vanadium atoms in 0.5 h- V_d - V_2O_3 and 6 h- V_d - V_2O_3 are 97.0(4)% (3.0% vanadium vacancies) and 95.3(1)% (4.7% vanadium vacancies), respectively, while no oxygen vacancies are found (**Supplementary Figs. 13 and 14** above). Furthermore, because of the engender of NH_3 and other gases in the hydrothermal process of vanadyl acetylacetonate and urea, the vacancy may be created during the hydrothermal process. Nevertheless, the V_2O_3 was not synthesized after hydrothermal treatment (**Fig. R2**), so it cannot be determined whether the vanadium vacancies were created, and the

vanadium vacancies concentration was difficult to control in the hydrothermal process. It can be concluded from the above results that the hydrothermal process may produce vacancies, while the calcination certainly creates vacancies, and the vacancy concentration can be controlled by changing the calcination time.

Fig. R2 | XRD patterns after hydrothermal.

4. Is the proposed method reliable to quantify the vanadium-vacancy in V_2O_3 ? The authors just assumed that c- V_2O_3 had no vanadium-vacancy. Can the authors prove that there is no vanadium-vacancy in c- V_2O_3 using the proposed method?

Reply: Thanks for this helpful suggestion.

Rietveld refinement with combined XRD and NPD patterns is a reliable method for accurate quantification of vanadium vacancies and oxygen vacancies. In the V_d - V_2O_3 cathode, the neutron scattering amplitude of the vanadium element is just -0.0382 cm^{-12} [Sears VF. Neutron-diffraction determination of magnetic structures. Transactions on Magnetics: 1972, 8: 161-182], and the atomic radius of the light oxygen element is about 0.66 \AA . As a result, the vanadium element in the V_d - V_2O_3 cathode is hard to be detected by NPD, while the light oxygen element is too small to be probed by XRD. However, the neutron scattering amplitude of the light oxygen element is as high as 0.5803 cm^{-12} , which is easy to be detected by NPD. The atomic radius of the vanadium

element is 1.22 Å, which is big enough to be probed by XRD. Accordingly, Rietveld refinement with combined XRD and NPD patterns is very accurate and reliable to reveal the crystal vacancy. Moreover, positron annihilation lifetime spectroscopy (PALS) and synchrotron X-ray analysis further confirmed the existence of vanadium vacancies.

To further prove the reliability of the proposed method, we used the Rietveld refinement with combined XRD and NPD patterns to prove that there are no vanadium vacancies in c-V₂O₃. As shown in **Supplementary Fig. 12**, the Rietveld analysis of c-V₂O₃ by combined XRD and NPD patterns reveals the crystal structure information. The refined results are displayed in **Supplementary Table 3** above. Like V_d-V₂O₃, c-V₂O₃ belongs to the identical space group, with similar lattice parameters and the positions occupied by V and O atoms. Nevertheless, the occupation ratios of vanadium and oxygen atoms are about 100%, indicating that there are no vanadium vacancies and oxygen vacancies in c-V₂O₃.

Supplementary Figure 12 | Neutron powder diffraction and X-ray diffraction refinement patterns for c-V₂O₃. Observed (black circle), calculated diffraction patterns (red line), their difference (purple line), and peak position (black bar) of the NPD pattern (upper part) and XRD pattern (lower part).

We have added these data and discussions in our revised Supplementary Information accordingly.

5. Electrochemical performance other than cycle stability of V_d-V₂O₃ should also be

compared with others.

Reply: Thanks to the reviewer's useful suggestion.

The energy density, power density, and rate capability have been compared with others ZIBs cathodes, as discussed following. **Supplementary Fig. 20** is the Ragone diagram of energy and power density of $V_d-V_2O_3$ electrode compared with other ZIBs cathodes. The results show that at a power density of 332.7 W kg^{-1} , the energy density of the $V_d-V_2O_3$ electrode is 110.9 Wh kg^{-1} , which is better than many reported ZIBs cathodes, such as $Na_3V_2(PO_4)_3$ [*Nano Energy*, 2016, 25, 211-217], $Na_{0.95}MnO_2$ [*Chem. Commun.*, 2014, 50, 1209-1211], $FeFe(CN)_6$ [*J Solid State Electrochem*, 2017, 21, 2021-2027], $CuHCF$ [*ChemSusChem*, 2015, 8, 481-485], $ZnHCF$ [*Adv. Energy Mater.*, 2015, 5, 1400930], VS_2 [*Adv. Energy Mater.*, 2017, 7, 1601920]. Furthermore, we compared the capacity retention rate of different ZIBs electrodes when the current density was expanded tenfold as shown in **Supplementary Fig. 19** and **Supplementary Table 4**. When the current density of the $V_d-V_2O_3$ electrode increases ten times, the capacity retention rate is still 70.4%, which is more remarkable than that of many ZIBs electrodes [*Energy Environ. Sci.*, 2021, 14, 3954-3964; *Adv. Energy Mater.*, 2019, 9, 1803815; *Nano Lett.*, 2018, 18, 1758-1763; *Adv. Funct. Mater.*, 2020, 30, 1907684; *Small Methods*, 2020, 4, 1900670].

Supplementary Figure 20 | Comparison of the Ragone plot of the $V_d-V_2O_3$ -based ZIB with other reported cathode materials.

Supplementary Figure 19 | Comparison of the capacity retention between $V_d-V_2O_3$ and other vanadium-based cathodes for aqueous ZIBs.

Supplementary Table 4 | Comparison of capacity retention for different vanadium-based aqueous ZIBs.

Sample	Current Density ($A\ g^{-1}$)	Capacity Retention Ratio(%)	Ref.
MnVO	0.2-2	50.0	Energy Environ. Sci. , 2021, 14, 3954-3964
O_d-MnO_2	0.2-2	51.4	Adv. Energy Mater. , 2019, 9, 1803815
$Na_2V_6O_{16} \cdot 1.63H_2O$	0.1-1	60.6	Nano Lett. , 2018, 18, 1758-1763
$Mn_{0.15}V_2O_5 \cdot nH_2O$	0.5-5	66.7	Adv. Funct. Mater. , 2020, 30, 1907684
C-KVO	0.2-2	68.9	Small Methods , 2020, 4, 1900670
$V_d-V_2O_3$	0.1-1	70.4	This work

We have added these data and discussions in our revised Supplementary Information accordingly.

6. The diffusion coefficient of Zn^{2+} in the $V_d-V_2O_3$ electrode is 10^{-7} to 10^{-8} cm^2/s , which is faster than other electrodes, including the above reference, but the rate capability of the $V_d-V_2O_3$ electrode is lower than that of the other electrodes. What would be the reason? Due to the low electrical conductivity of the $V_d-V_2O_3$ electrode?

Reply: Thanks for your helpful suggestion.

The limitation to the rate capability can arise from numerous sources including an

increase of the ohmic contribution (active material resistance, solid-electrolyte interphase resistance), diffusion limitations [*Nat. Mater.*, 2013, 12, 518-522; *J. Power Sources*, 2010, 195, 7904-792], phase transformation kinetics, enlarge of particle size, and the reduction of electronic conductivity of the materials [*J. Power Sources*, 2007, 164, 849-856].

V_d-V₂O₃ and c-V₂O₃ were pressed into thin blocks respectively, and then the material resistance was measured by the four-probe method. At room temperature (300K), the resistivity of V_d-V₂O₃ and c-V₂O₃ are 832 Ω cm and 2 Ω cm respectively, and the corresponding conductivity is 0.001 S cm⁻¹ and 0.5 S cm⁻¹. It can be seen that although V_d-V₂O₃ has a higher diffusion coefficient of Zn²⁺, its electrical conductivity is lower as expected by the reviewer, bringing out the mediocre rate capability of the electrode.

Reviewer #2 (Remarks to the Author):

The paper deals with an important topic related to improving the performance of rechargeable aqueous Zinc ion batteries through the structural defects in cathode materials. The authors investigated vanadium defect (V_d) clusters in the V_2O_3 lattice by combining spectroscopy analytical techniques (Neutron powder diffraction (NPD), X-ray powder diffraction (XRD), positron annihilation spectroscopy (PAS) and synchrotron-based X-ray) and theoretical calculations (Rietveld refinement and density functional theory (DFT) calculation). The authors demonstrated that V_d clusters can provide permanent sites for Zn^{2+} anchormen to enhance the integrity of V_2O_3 after the first discharging process. It also make Zn^{2+} de/intercalation in complex oxide, which contributes in reducing the strong electrostatic interaction between host multivalent ions. They authors correlate the V_d defect clusters with the with ultra-long cycle life of V_d - V_2O_3 cathode for Zn^{2+} storage. A full complete set of experimental data and theoretical calculations was provided with good interpretation of these results.

The reviewer finds that the novelty of the work in quantifying the structural defects in cathode material and the engineering design of the V_d - V_2O_3 cathode material is unique in this work. The paper can be accepted for publication in nature communications after considering the following changes:

We sincerely appreciate the reviewer for the positive evaluations and the detailed suggestions on our work. The following comments have been addressed point-by-point, and the manuscript has been revised accordingly.

1-Abstract.

The following sentence “Besides, the density functional theory (DFT) calculations storage performance of Zn^{2+} ” is very long. Please split in 2-3 sentences for clarification.

Reply: Thank you for your helpful advice. We have revised this sentence in the manuscript to: “The density functional theory (DFT) calculations strongly indicate that the V_d clusters provide permanent sites for Zn^{2+} anchormen to enhance the integrity of V_2O_3 after the first discharging process. Furthermore, the V_d clusters make Zn^{2+}

de/intercalation in complex oxide, contributing collectively and effectively reducing the strong electrostatic interaction between host multivalent ions, resulting in the remarkable storage performance of Zn^{2+} .”

2- Introduction.

(1) “... security issues...”: Please change the word “security” with more appropriate word (“safety” or “hazard”).

(2) “... ZIBs have great commercial potential. But great upgrading ...”: The word “great” is used consecutively twice. Please replace one of the words “great” with another word having the same meaning (important, huge, suitable, ...).

(3) “Accurately quantification ...”: Change “Accurately” with “Accurate”

Reply: Thank you for your useful suggestions. We have revised the above questions in the manuscript accordingly.

(4) “For example, light oxygen is difficult to be detected by XRD owing to its small atomic radius, compared with most metal elements.”: Please support this idea with references (example: Chem. Sci.,2021,12,517-532).

Reply: Thank you for your useful suggestions. We have added book [The basics of cristallography and diffraction. Vol. 214 (Oxford, 2001)] in the manuscript accordingly.

(5) “For example, light oxygen is difficult to be detected by XRD owing to its small atomic radius, compared with most metal elements. Consequently, the joint application of multiple spectroscopy is essential to accurately quantify defects.”: The transition between ideas is not evident. Please revise part to clearly state the novelty of your work.

Reply: Thank you for your useful suggestions. We have revised this sentence in the manuscript to: “For example, Rietveld refinement of XRD is commonly used to quantify defect concentration. Nevertheless, light oxygen is difficult to be detected by XRD owing to its small atomic radius, compared with most metal elements. Rietveld refinement of NPD is also often used to quantify defect concentration. However, the scattering factor of heavier elements is too small to be probed by NPD. Consequently,

the joint application of multiple spectroscopy is essential to accurately quantify defects.”

3. Structure and morphology characterization of V_d - V_2O_3

(1) “But the situation in V_d - V_2O_3 is different, the surface of V_d - V_2O_3 is proved to be uniformly coated with carbon”: The reviewer did not notice a difference in carbon content in both V_d - V_2O_3 c- V_2O_3 materials? For c- V_2O_3 material, the carbon content should be provided from the supplier if it exists. The TGA analysis of c- V_2O_3 material did not show that carbon exist in its composition. Please clarify this result. Can the carbon content be controlled during hydrothermal treatment and annealing? Do the author use any other source of vanadium for comparison? If yes, please provide the data of mapping analysis.

Reply: Thank you for your helpful advice.

(1) According to TGA curve data, we calculated that the carbon content of c- V_2O_3 is only about 1% (The supplier claimed that the carbon content did not be measured.), while that of V_d - V_2O_3 is 22.92%. Furthermore, Raman spectroscopy further confirmed the presence of carbon in V_d - V_2O_3 , but not in c- V_2O_3 (**Fig. R1a**). The SEM element mapping images of c- V_2O_3 display a small amount of carbon, which may be attributed to the small amount of carbon in the air adsorbed on the surface of c- V_2O_3 .

(2) By controlling different hydrothermal treatment and annealing conditions, we found that changing the hydrothermal temperature and time (in the case of other conditions unchanged, the hydrothermal temperature changed to 180 °C and 190 °C respectively, the hydrothermal time changed to 10 h and 12 h respectively), which will change the carbon content, but the variation is small (**Fig. R1b**). Changing the calcination time to 400 °C, 500 °C, 700 °C, and 800 °C respectively, it is found that V_2O_3 cannot be synthesized at 400 °C and 500 °C (**Figs. R1c and R1d**). The calcination temperature has a great influence on V_2O_3 , and the carbon content decreases with the increase of calcination temperature (**Fig. R1b**).

(3) We also used other vanadium sources for comparison. 1 mmol ammonium vanadate and 1.5 mmol hydroxylamine hydrochloride were mixed into 40 mL ethylene glycol and stirred until the solution is transparent. Then put it into a Teflon-lined sealed

autoclave (50 ml) and keep it at 180 °C for 18 h. After centrifugation, washing, and cold drying, the powder was heated in an N₂ atmosphere at 600 °C for 3 h. According to the TGA curve, the carbon content of V₂O₃ synthesized by this method is very small, about 1% (**Fig. R1b**). Nevertheless, the SEM element mapping images (**Fig. R2**) show that the presence of carbon element, which further confirms the conjecture that it may be the adsorption of carbon in the air.

Fig. R1 | **a**, Raman spectra of the V_d-V₂O₃ and c-V₂O₃. **b**, TGA curves of samples treated by different hydrothermal treatment and annealing process. **c**, XRD patterns at 400 °C and 500 °C calcination temperature. **d**, XRD patterns at 700 °C and 800 °C calcination temperature.

Fig. R2 | SEM element mapping images of other vanadium sources.

(2) Are the provided data in Figure Supplementary Figure 9 are the best fitting for the Fourier transformed EXAFS? The authors suggested that CN less than 3 for V in V_d - V_2O_3 to V vacancies. Please support your suggestion with references from literature reported for V or any other transition metal oxides.

Reply: Thank you for your useful suggestion. The provided data in **Supplementary Fig. 10** are the best fitting for the Fourier transformed EXAFS (Supplementary Figure 9 has been revised to Supplementary Figure 10). We have added references [*J. Am. Chem. Soc.*, 2016, 138, 6517-6524; *ACS Catal.*, 2018, 8, 3803-3811; *Energy Stor. Mater.*, 2020, 24, 394-401] in the manuscript accordingly.

Supplementary Figure 10 | Experiment and fitted curves of k^3 weighted Fourier transform for $V_d-V_2O_3$ (a) and $c-V_2O_3$ (b).

(3) PAS analysis results can be more detailed. $V_d-V_2O_3$ and $c-V_2O_3$ showed similar I_3 intensity. However, differences can be observed in I_1 and I_2 intensities between $V_d-V_2O_3$ and $c-V_2O_3$. The authors focused only on I_2 . Please provide a full interpretation of the PAS results.

Reply: Thank you very much for your reminder.

To further corroborate the defect situation in $V_d-V_2O_3$, we used positron annihilation lifetime spectroscopy (PALS) to explore the defect type and concentration of the material [Principles and Applications of Positron and Positronium Chemistry. (World Scientific, 2003)]. **Table 1** shows the PALS results of $V_d-V_2O_3$ and $c-V_2O_3$. The PALS spectra were well decomposed into three life components (τ_1 , τ_2 , and τ_3). The shortest lifetime component (τ_1) corresponds to the positron annihilation in the defect-free bulk regions and tiny vacancies. The longer lifetime component (τ_2) is attributed to the positron annihilation in vacancy clusters or boundary regions. Combined with the following Rietveld refinement with combined XRD and NPD patterns, the τ_2 of $c-V_2O_3$ probably originates from positron annihilation in the boundary regions [*J. Mater. Sci.*, 1999, 34, 3833-3851]. The longest component (τ_3) of several nanoseconds, which results from the annihilation of ortho-positronium (boundary state of a positron and an electron, spin triplet) in some large voids which is the unoccupied space among close-packed nanograins, is not correlated with the microstructure in nanograins. Accordingly,

we could neglect the longest component. The values of τ_1 (175 ps), τ_2 (392 ps), and τ_3 (1.610 ns) in c-V₂O₃ are very close to the previous results (171 ps, 414 ps, and 1.8 ns) [*J. Phys. Soc. Jpn.*, 1973, 34, 661-665]. For the V_d-V₂O₃ sample, the shortest lifetime component is obviously lower than that of c-V₂O₃, indicating the significant existence of vacancy-type defects in V_d-V₂O₃, while obvious large vacancy-type defects do not exist in c-V₂O₃. It is noteworthy that the intensity I_2 of τ_2 for V_d-V₂O₃ is 78.01%, which is much higher than that of c-V₂O₃, further confirmed that the concentration of vacancy clusters in V_d-V₂O₃ is much higher than that in c-V₂O₃.

We have added these discussions in our revised manuscript accordingly.

(4) Consistent abbreviation is required: Use PAS or PALS?

Reply: Thank you for your helpful advice. The consistent abbreviation should be PALS (positron annihilation lifetime spectroscopy). We have revised in the manuscript accordingly.

(5) Please provide NPD and XRD spectra of c-V₂O₃ (as supplementary material) to observe the differences with V_d-V₂O₃.

Reply: Thanks for this helpful suggestion.

As shown in **Supplementary Fig. 12**, the Rietveld analysis of c-V₂O₃ by combined X-ray powder diffraction (XRD) and Neutron powder diffraction (NPD) patterns reveals the crystal structure information. The refined results are illustrated in **Supplementary Table 3**. Like V_d-V₂O₃, c-V₂O₃ belongs to the identical space group, with similar lattice parameters and the positions occupied by V and O atoms. Nevertheless, the occupation ratios of vanadium and oxygen atoms are about 100%, indicating that there are no vanadium vacancies and oxygen vacancies in c-V₂O₃.

Supplementary Figure 12 | Neutron powder diffraction and X-ray diffraction refinement patterns for $c\text{-V}_2\text{O}_3$. Observed (black circle), calculated diffraction patterns (red line), their difference (purple line), and peak position (black bar) of the NPD pattern (upper part) and XRD pattern (lower part).

Supplementary Table 3 | Structural parameters from Rietveld refinement for $V_d\text{-V}_2\text{O}_3$.

Sample	Lattice parameters			Atomic occupancy		wRp	χ^2
$V_d\text{-V}_2\text{O}_3$	$a=4.94734(9) \text{ \AA}$	$\alpha=90^\circ$	$V=0.296$ 74 nm^3	V	O	wRp(NDP) =2.95%	22.5
	$b=4.94734(9) \text{ \AA}$	$\beta=90^\circ$		94.3(1)%	102.7(3)% $\approx 100\%$	wRp(XRD) =3.6%	
	$c=13.9990(5) \text{ \AA}$	$\gamma=120^\circ$					
$c\text{-V}_2\text{O}_3$	$a=4.95747(9) \text{ \AA}$	$\alpha=90^\circ$	$V=0.298$ 26 nm^3	V	O	wRp(NDP) =3.76%	5.89
	$b=4.95747(9) \text{ \AA}$	$\beta=90^\circ$		100.7(6)%	99.9(1)%	wRp(XRD) =13.77%	
	$c=14.01323(8) \text{ \AA}$	$\gamma=120^\circ$					
$0.5 \text{ h-}V_d\text{-V}_2\text{O}_3$	$a=4.95594(7) \text{ \AA}$	$\alpha=90^\circ$	$V=0.298$ 20 nm^3	V	O	wRp(NDP) =3.58%	2.86

	b=4.95594(7) Å	$\beta=90^\circ$			97.0(4)%	100.5(1)%	wRp(XRD) =12.55%	
	c=14.0192(4) Å	$\gamma=120^\circ$						
6 h-V _d - V ₂ O ₃	a=4.95720(6) Å	$\alpha=90^\circ$	V=0.298 40 nm ³	V	O		wRp(NDP) =3.71%	2.49
	b=4.95720(6) Å	$\beta=90^\circ$						
	c=14.0214(3) Å	$\gamma=120^\circ$		95.3(1)%	100.0(3)%	wRp(XRD) =10.72%		

We have added these data and discussions to our revised manuscript accordingly.

(6) “In general, we have demonstrated the presence of coordinately unsaturated atoms of V in V_d-V₂O₃ and the vacancy occupancy rate of vanadium is 5.7%, and the vanadium vacancies exist in the form of vacancy clusters.” : Please revise this sentence to make a reasonable conclusion.

Reply: Thank you for your useful suggestions. We have revised this sentence in the manuscript to: “Accordingly, it can be concluded that the V_d-V₂O₃, 0.5 h-V_d-V₂O₃, and 6 h-V_d-V₂O₃ contain 5.7%, 3.0%, and 4.7% vanadium vacancies respectively, no oxygen vacancies exist, while c-V₂O₃ has neither vanadium vacancies nor oxygen vacancies.”

4. Electrochemistry

(1) The authors used both ZnSO₄ and Zn(CF₃SO₃)₂ supporting electrolytes in their electrochemical tests. One sentence should be added to explain the differences between them and the effect of the concentration of the supporting electrolyte.

Reply: Thank you very much for your useful suggestion.

Mild acid electrolytes such as ZnSO₄ solution [*J. Chem. Mater.*, 2015, 27, 3609-3620; *Chem. Commun.*, 2015, 51, 9265-9268] have inherently limited solubility and coulomb efficiency of Zn stripping/plating, while Zn(CF₃SO₃)₂ solution has high ionic

conductivity and electrochemical stability [*J. Am. Chem. Soc.*, 2016, 138, 12894-12901; *Adv. Mater.*, 2018, 30, 1800762], and is widely used in aqueous ZIBs. Moreover, higher salt concentration can reduce the water activity and side reaction caused by water [*Science*, 2015, 350, 938-943; *Energy Environ. Sci.*, 2016, 9, 1841-1848], thus improving the cycling stability of the electrode in aqueous solution.

We have added these discussions in our revised manuscript accordingly.

(2) The specific capacity of V_d - V_2O_3 is lower than other cathode materials reported in the literature. This should be clearly stated.

Reply: Thanks for pointing out this issue.

The statement about the specific capacity of V_d - V_2O_3 being lower than other cathode materials reported in the literature is separated into two parts, namely the V_2O_3 -based electrodes and the vanadium-based oxide electrodes.

(1) The results for the V_d - V_2O_3 show a lower capacity than other V_2O_3 -based electrodes mainly attribute to the following three points.

First of all, the voltage window is wider than the voltage window in our manuscript, resulting in higher capacity. For example, the voltage window of the references (*ACS Nano* 2020, 14, 6, 7328-7337; *Small Methods* 2021, 2100578) is measured within the range of 0.2-1.6 V. Accordingly, as a comparison, we expanded the test voltage to 1.6 V and found that charging curves similar to that in the references appeared during the first cycle of charging. After the first cycle of charging, the V_2O_3 capacity increased sharply from 187 mA h g⁻¹ at 0.1 A g⁻¹ to 422 mA h g⁻¹ (**Fig. R3a**). Furthermore, the CV curves also arose similar oxidation peaks at the first cycle of charging to that in the references (**Fig. R3b**). Nevertheless, the oxidation of V_d - V_2O_3 at 1.6 V voltage will bring about structural changes (**Fig. R3c**), and the vanish of vanadium vacancies, which cannot provide excellent stability. After less than 1800 cycles, the capacity retention rate is only 86.2% (**Fig. R3d**). The reference (*ACS Nano* 2020, 14, 6, 7328-7337) also compared the charging to 1.3 V and 1.6 V respectively, and it is found that the capacity does change greatly. Consequently, we did not expand the voltage window to 1.6 V. Secondly, the electrode material undergoes a phase change during the cycle. For

instance, the electrode material is converted from V_2O_3 material to $V_2O_{5-x} \cdot nH_2O$ after the first cycle of anodic oxidation in the reference (*ACS Nano* 2020, 14, 6, 7328-7337), which is actually the capacity of $V_2O_{5-x} \cdot nH_2O$. Furthermore, the V_2O_3 phase disappears after the initial charge process, and $Zn_{3+x}(OH)_{2+3x}V_{2-x}O_{7-3x} \cdot 2H_2O$ and zinc vanadate (Zn_yVO_z) phases arise from the second cycle in the reference (*Small Methods* 2021, 2100578). However, the structure of our V_2O_3 electrode has not changed after many cycles (**Supplementary Fig. 27**). Last but not least, the reference (*ACS Nano* 2020, 14, 6, 7328-7337) stated that carbon materials are unsuitable for the storage of Zn^{2+} , and the carbon content was excluded in the calculation of the capacity. Nevertheless, we include the carbon content in the calculation, which will also cause the V_d - V_2O_3 electrode to have a lower capacity than the reference.

(2) Compared with many other vanadium-based oxide electrodes [*Sci. Adv.*, 2019, 5, eaax4279; *Energy Environ. Sci.*, 2021, 14, 4095-4106; *Energy Environ. Sci.*, 2019, 12, 2273-2285; *Angew. Chem. Int. Ed.*, 2019, 58, 7062-7067; *Adv. Mater.*, 2018, 30, 1703725; *Adv. Mater.*, 2018, 30, 1705580], the voltage window is wider than the V_d - V_2O_3 electrode bringing about higher capacity. Moreover, the voltage of these vanadium-based oxide electrodes is not less than 1.6 V, while the V_d - V_2O_3 electrode will oxidize at 1.6 V, resulting in the disappearance of vanadium vacancies and structural changes, which will affect its ultra-long cycle stability.

Fig. R3 | **a**, Rate performance at different current densities of $V_d-V_2O_3$ electrode in 0.1-1.6 V voltage window. **b**, CV curves of V_2O_3 in $Zn(CF_3SO_3)_2$ electrolyte at $0.1 mV s^{-1}$. **c**, XRD pattern of the $V_d-V_2O_3$ electrode charged to 1.6 V after 10 cycles. **d**, Long-term cycling performance and coulombic efficiency of $V_d-V_2O_3$ electrode at a current density of $5 A g^{-1}$ in 0.1-1.6 V voltage window.

Supplementary Figure 27 | XRD patterns of $V_d-V_2O_3$ electrodes under different cycles.

(3) “The longevity of this V_d - V_2O_3 cathode comes ultimately from abundant vacancy clusters that attenuate the strong electrostatic interaction between Zn^{2+} and the V_d - V_2O_3 .”: This conclusion is not well placed in the manuscript. It should be moved to the end.

Reply: Thank you for your useful advice. We have moved the conclusion to the end in the manuscript accordingly.

(4) “the c- V_2O_3 cathode, without vacancies, ...”: This is wrong based on PAS results. Please revise the sentence for accuracy.

Reply: Thank you for your useful advice. We have revised this sentence in the manuscript to: “the 0.5 h- V_d - V_2O_3 (3.0% vanadium vacancies), 6 h- V_d - V_2O_3 (4.7% vanadium vacancies), and c- V_2O_3 (no vanadium vacancies) cathodes demonstrate inferior both rate and stability electrochemical performance for aqueous ZIBs, which strongly confirms the positive effects of vanadium vacancies in V_d - V_2O_3 cathode and 5.7% vanadium vacancies has the most excellent electrochemical performance, especially cycle stability.”

5- Zinc-ion storage mechanism of V_d - V_2O_3

(1) Can the authors provide a clear explanation of the difference in V defects between the DFT model (6.25%) and experimental data (5.7%)?

Reply: Thanks for the useful suggestions.

In the DFT calculation, a model containing 32 V atoms and 48 O atoms. Consequently, the concentration of V vacancy can be set as $1/32 \times 100\% = 3.125\%$, $2/32 \times 100\% = 6.25\%$, and so on. To simulate realistic experimental data (5.7%), the concentration of V vacancy was set as 6.25%.

(2) Please number the structures in Supplementary Figure 18.

Reply: Thank you for your useful proposition. We have numbered the structures in **Supplementary Fig. 26** accordingly (Supplementary Figure 18 has been revised to

Supplementary Figure 26).

Supplementary Figure 26 | six random structure models of different vanadium vacancies in $V_d-V_2O_3$ at the concentration of 6.25%.

(3) The abbreviation p- V_2O_3 should be provided the first time is cited in the text.

(4) “a large number of heat ...”: Please change “number” with “amount” or “quantity”

(5) “When Zn^{2+} initially entries ...”: Please replace “entries” with “enters”

Reply: Thank you for your useful suggestions. We have revised the above questions in the manuscript accordingly.

(6) “the eventual structure is a Zn doped $V_d-V_2O_3$ after the first discharging self-optimized process, in which the Zn^{2+} will reversibly insert or leave in the subsequent cycles.”: Please correlate this suggestion with the initial coulombic efficiency of Zn/ $V_d-V_2O_3$ cell.

Reply: Thanks for this helpful suggestion.

Based on your very useful suggestions, we compared the Coulomb efficiency of the first cycle and the second cycle, which were 87.1% and 93.6% respectively. This further confirms that the $V_d-V_2O_3$ electrode undergoes a self-optimization process after the first discharge.

We have mentioned this discussion in the revised manuscript accordingly.

Reviewer #3 (Remarks to the Author):

Comments:

I have reviewed the article titled “Quantifying vanadium-vacancy clusters in V_2O_3 towards ultra-long cycling aqueous zinc-ion battery”. The authors have carried out exhaustive spectroscopy and diffraction studies in order to establish the presence of vanadium cluster vacancies in the V_2O_3 cathode material.

We highly appreciate the reviewer for his/her positive evaluation and careful suggestions on our work. Accordingly, we have revised the manuscript and would like to reply the reviewer’s concerns one-by-one as below.

Although many studies were done, there is no categorical proof that the presence of vacancy is what gave rise to capacity and good cycle life for the zinc-ion battery. The authors fail to establish a strong correlation between vacancy and battery performance, they should have made a series of V_2O_3 having varying degree of vacancy and correlated its effect to the battery performance. Since a number of studies have been done in this work, it should have been submitted as a full manuscript instead of communication. In order to make it look like a communication, authors have curtailed the discussion part significantly without providing crucial information.

Reply: Thank you for your very useful advice.

According to the experimental and DFT calculations, we find that the vanadium vacancies can effectively promote the cycle stability of ZIB, and make it exhibit ultra-long cycle stability. The details are as follows: In the manuscript, the performances of the V_d - V_2O_3 (5.7% vanadium vacancies) electrode and the c - V_2O_3 (no vanadium vacancies) electrode are compared. It is found that 5.7% vanadium vacancies make V_d - V_2O_3 electrodes have excellent cycle stability. Moreover, DFT calculation further proved that part of vanadium vacancies provides permanent sites for the preoccupation of a small amount of Zn^{2+} so that the system will have a more stable structure to against collapsing during the process of Zn^{2+} insertion/extraction. Meanwhile, the other vanadium vacancies can effectively weaken the strong interaction between Zn^{2+} and the V_2O_3 material host to allow free insertion/extraction of Zn^{2+} . Last but not least, through

X-ray powder diffraction (XRD), soft X-ray absorption spectrum (sXAS), and X-ray photoelectron spectroscopy (XPS), we have further certified that the structure and valence changes of the $V_d-V_2O_3$ electrode are extremely reversible during the charging and discharging process, which reflects the stability of $V_d-V_2O_3$.

As suggested, in order to establish the relationship between vacancy and battery performance, we have controlled the vanadium vacancies concentration by controlling the calcination time. Consequently, we did a set of comparative experiments, under the same other conditions, alter the calcination time for 0.5 h and 6 h (0.5 h- $V_d-V_2O_3$ and 6 h- $V_d-V_2O_3$) respectively. The structure information of 0.5 h- $V_d-V_2O_3$ and 6 h- $V_d-V_2O_3$ was revealed by Rietveld refinement with combined X-ray powder diffraction (XRD) and Neutron powder diffraction (NPD) patterns. The refined results are shown in **Supplementary Table 3**, which displays that the structures of 0.5 h- $V_d-V_2O_3$ and 6 h- $V_d-V_2O_3$ are the same as the $V_d-V_2O_3$. Nevertheless, the occupation ratios of 0.5 h- $V_d-V_2O_3$ and 6 h- $V_d-V_2O_3$ are 97.0(4)% (3.0% vanadium vacancies) and 95.3(1)% (4.7% vanadium vacancies), respectively, while no oxygen vacancies are found (**Supplementary Figs. 13 and 14**), indicating that the vanadium vacancies concentration at both calcination temperatures is less than that of $V_d-V_2O_3$. We made the V_2O_3 at these two calcination temperatures into ZIB electrodes, both of which have inferior rate performance and cycle stability (**Supplementary Fig. 22**). On the basis of DFT calculation, we can conclude that the $V_d-V_2O_3$ electrode with vanadium vacancies concentration of 5.7% has the most excellent electrochemical performance, especially cycle stability (**Supplementary Table 6**). The manuscript includes the synthesis, characterization, electrochemical application, and mechanism of the $V_d-V_2O_3$, and many discussions are made. Consequently, we submitted it as a full manuscript. Furthermore, we have added many discussions in our revised manuscript to make it look more like a complete manuscript.

Supplementary Table 3 | Structural parameters from Rietveld refinement for V_d - V_2O_3 .

Sample	Lattice parameters			Atomic occupancy		wRp	χ^2
V_d - V_2O_3	a=4.94734(9) Å	$\alpha=90^\circ$	V=0.296 74 nm ³	V	O	wRp(NDP) =2.95%	22.5
	b=4.94734(9) Å	$\beta=90^\circ$		94.3(1)%	102.7(3)% $\approx 100\%$	wRp(XRD) =3.6%	
	c=13.9990(5) Å	$\gamma=120^\circ$					
c- V_2O_3	a=4.95747(9) Å	$\alpha=90^\circ$	V=0.298 26 nm ³	V	O	wRp(NDP) =3.76%	5.89
	b=4.95747(9) Å	$\beta=90^\circ$		100.7(6)%	99.9(1)%	wRp(XRD) =13.77%	
	c=14.01323(8) Å	$\gamma=120^\circ$					
0.5 h- V_d - V_2O_3	a=4.95594(7) Å	$\alpha=90^\circ$	V=0.298 20 nm ³	V	O	wRp(NDP) =3.58%	2.86
	b=4.95594(7) Å	$\beta=90^\circ$		97.0(4)%	100.5(1)%	wRp(XRD) =12.55%	
	c=14.0192(4) Å	$\gamma=120^\circ$					
6 h- V_d - V_2O_3	a=4.95720(6) Å	$\alpha=90^\circ$	V=0.298 40 nm ³	V	O	wRp(NDP) =3.71%	2.49
	b=4.95720(6) Å	$\beta=90^\circ$		95.3(1)%	100.0(3)%	wRp(XRD) =10.72%	
	c=14.0214(3) Å	$\gamma=120^\circ$					

Supplementary Figure 13 | Neutron powder diffraction and X-ray diffraction refinement patterns for 0.5 h-V_d-V₂O₃. Observed (black circle), calculated diffraction patterns (red line), their difference (purple line), and peak position (black bar) of the NPD pattern (upper part) and XRD pattern (lower part).

Supplementary Figure 14 | Neutron powder diffraction and X-ray diffraction refinement patterns for 6 h-V_d-V₂O₃. Observed (black circle), calculated diffraction patterns (red line), their difference (purple line), and peak position (black bar) of the NPD pattern (upper part) and XRD pattern (lower part).

Supplementary Figure 22 | a, Rate performance at different current densities of 0.5 h-V_d-V₂O₃ electrode. **b,** Rate performance at different current densities of 6 h-V_d-V₂O₃ electrode. **c,** Cycling performance and coulombic efficiency of 0.5 h-V_d-V₂O₃ electrode at a current density of 5 A g⁻¹. **d,** Cycling performance and coulombic efficiency of 6 h-V_d-V₂O₃ electrode at a current density of 5 A g⁻¹.

Supplementary Table 6 | Performance comparison of V₂O₃ under different vanadium vacancies concentrations

Sample	Vanadium vacancies concentrations	Specific capacity at 0.3 A g ⁻¹	Number of cycles at 5 A g ⁻¹	Retention rate after cycle
V _d -V ₂ O ₃	5.7%	187 mA h g ⁻¹	30,000	81.0%
0.5 h-V _d -V ₂ O ₃	4.7%	150 mA h g ⁻¹	600	44.0%
6 h-V _d -V ₂ O ₃	3.0%	184 mA h g ⁻¹	1000	42.8%
c-V ₂ O ₃	0.0%	18 mA h g ⁻¹	150	76.9%

Moreover, the performance of the battery is not very impressive, I could find even organic cathode materials with similar capacity and able to operate upto 63C rate (ACS Appl. Energy Mater. 2021, 4, 1218-1227). My specific comments are given below.

Reply: Thanks for the helpful suggestion.

We compared the differences between the two articles and found that they have analogous capacity. Nevertheless, there are great disparities in the highlights of the articles. In the mentioned reference, PT/Super P electrodes showed poor rate capacity without additive CMK-3. In fact, the reference mainly focuses on highlighting the effect of additives on improving rate performance, while the brightened dot of our work is that 5.7% of V_d clusters in V_d - V_2O_3 cathode are quantified for the first time in aqueous ZIBs, which shows remarkable Zn^{2+} storage performance. It is the V_d clusters that make V_d - V_2O_3 cathode have excellent cycle stability, which is the longest aqueous ZIBs stability as far as we know.

Accordingly, we have emphasized this difference and stated the long-cycling significance of our work in the revised manuscript.

1. Replace “GTA” with “TGA” in the caption of supplementary Fig. 8

Reply: Thanks, we apologize for the mistake in expression. We have corrected **Supplementary Fig. 9** in the revised manuscript (Supplementary Fig. 8 has been revised to Supplementary Fig. 9).

Supplementary Figure 9 | TGA curves of V_d-V₂O₃ and c-V₂O₃.

2. TGA of c-V₂O₃ shows an increase in wt. % up to 19.8%, whereas it is stated otherwise in the text. Why would the wt.% increase with the increase in temperature? It is not clear whether it is performed in an oxygen environment or something else.

Reply: Thanks for the useful suggestion.

We have mentioned in the supplementary information that TGA was carried out at 10 °C min⁻¹ on Q5000 in an air atmosphere. It is attributed to the oxidation of c-V₂O₃ to V₂O₅ (V₂O₃ + O₂ → V₂O₅) in the presence of O₂ that the wt.% in the TGA curve of c-V₂O₃ increases with the increase of temperature. If the samples are all V₂O₃ and completely oxidized to V₂O₅, the wt.% should be increased by 21.3%. Nevertheless, owing to a small amount of carbon in c-V₂O₃, carbon reacts with oxygen (C + O₂ → CO₂), bringing about the wt.% of 19.84%. Consequently, the content of carbon in c-V₂O₃ is about 1.2%.

3. It is stated by authors that “carbon on V_d-V₂O₃ is not easily oxidized”, but the data shows a drop in wt.% with temperature, hence the statement is not matching with the figure.

Reply: Thank you very much for kind reminder.

We have revised it into a more rigorous statement in the manuscript: “Surface

coated with the carbon of $V_d-V_2O_3$ is believed to be not easily oxidized at room temperature, so the higher content of V^{4+} in $V_d-V_2O_3$ may attribute to the existence of vanadium vacancies, which leads to a valence increase of V.” The change of the TGA curve of $V_d-V_2O_3$ is on account of these two reactions in the presence of oxygen:

Reaction 1 makes wt.% increase with the increase of temperature, while reaction 2 is the contrary. According to the TGA curve of c- V_2O_3 , reaction 1 occurred at 300-500 °C, while reaction 2 arose at the beginning of the decrease of wt.% in the TGA curve of $V_d-V_2O_3$. Furthermore, the wt.% of c- V_2O_3 and $V_d-V_2O_3$ TGA curves did not change after 600 °C, indicating that the reaction was complete (**Supplementary Fig. 9** above). Consequently, the carbon content in $V_d-V_2O_3$ can be calculated based on the final wt.%. The carbon content of c- V_2O_3 and $V_d-V_2O_3$ is 1.2% and 22.92% respectively.

4. Fitting in Fig. 9 is not satisfactory beyond 3 Å, is there any specific reason?

Reply: Thanks for the useful suggestion.

In the process of fitting, beyond 3 Å in the R space is too complicated attribute to multiple scattering of X-ray and other reasons, so we didn't fit it. Furthermore, we mainly discuss the V-V bond of $V_d-V_2O_3$ and c- V_2O_3 in the first shell and the V-O bond of the second shell (**Supplementary Fig. 9** has been revised to **Supplementary Fig. 10**). It is dispensable to research beyond 3 Å for this material.

5. It is indicated the presence of two pairs of redox peaks located at 1.09/0.78 V and 0.93/0.53V, which is not matching with supplementary Fig. 13 and redox pairs are not clear. I see redox pairs at 0.7/0.55 and 1.1/0.9 V.

Reply: Thanks for the helpful suggestion.

We apologize for the incorrect statements about two pairs of the position order of redox peaks. The two pairs of redox peaks in **Supplementary Fig. 17** should be located at 1.09/0.93 V and 0.78/0.53 V, respectively (**Supplementary Fig. 13** has been revised to **Supplementary Fig. 17**). We have revised it in the manuscript.

Supplementary Figure 17 | CV curves of the 1st, 2nd, and 3rd cycles for V_d-V₂O₃ electrode at a scan rate of 0.1 mV s⁻¹.

6. In supplementary fig 13, the area under the first cycle is lower than the later two cycles. How reproducible is this data? Why would the area increase with cycle? Do authors repeat this experiment and confirm the results?

Reply: Thanks for the useful advice.

The repeatability of **Supplementary Fig. 17** above is pretty good in our experiments (Supplementary Fig. 13 has been revised to Supplementary Fig. 17). We repeated it many times for long period, which is still this result. As shown in **Fig. R1**, the CV curves of the other V_d-V₂O₃ electrodes are almost the same as the **Supplementary Fig. 17** above.

Fig. R1 | **a**, CV curves of the 1st, 2nd, and 3rd cycles for the 1[#] V_d-V₂O₃ electrode at a scan rate of 0.1 mV s⁻¹. **b**, CV curves of the 1st, 2nd, and 3rd cycles for the 2[#] V_d-V₂O₃ electrode at a scan rate of 0.1 mV s⁻¹.

With the progress of the cycles, the area increases gradually, which may be attributed to the continuous activation of the V_d-V₂O₃ electrode in the first few cycles, resulting in the continuous increase of the area and the corresponding capacity [*J. Am. Chem. Soc.*, 2016, 138, 12894-12901; *Nat. Chem.*, 2012, 4, 579-585; *J. Mater. Chem. A*, 2018, 6, 23757-23765; *J. Mater. Chem. A*, 2017, 5, 17990-17997; *Energy Stor. Mater.*, 2018, 15, 374-379]. This phenomenon is also consistent with the continuous increase of the specific capacity in the first few cycles in **Fig. 2a**. Furthermore, we also compared the CV curves (**Fig. R2**) of the c-V₂O₃ electrode and found that the area of the c-V₂O₃ electrode also gradually increased in the first few cycles, which further proved our conjecture.

Fig. 2 | a, Rate performance at different current densities of $V_d-V_2O_3$ electrode.

Fig. R2 | CV curves of the 1st, 2nd, and 3rd cycles for $c-V_2O_3$ electrode at a scan rate of 0.1mV s^{-1} .

7. Does $V_d-V_2O_3$ undergoes any surface change due to cycling? Any in-situ evidence?

Reply: Thanks for your useful suggestion.

The structure and morphology of the $V_d-V_2O_3$ electrode are unchanged during the cycling process, while the valence state of the electrode surface would periodically change. We are awfully sorry that there are currently no available in-situ tools from our side to observe the changes of the $V_d-V_2O_3$ electrode surface with the cycle.

Nevertheless, a series of ex-situ characterizations were used to observe the surface evolution of the V_d - V_2O_3 electrode with cycling. The detailed discussion is as follows.

(1) **Surface morphology of V_d - V_2O_3 electrode:** We observed by ex-situ SEM that the surface morphology of the V_d - V_2O_3 electrode did not change under the different number of scanning cycles and different voltages (**Supplementary Figs. 28-30**).

Supplementary Figure 28 | SEM images of V_d - V_2O_3 electrodes at different voltages in 1st circle.

Supplementary Figure 29 | SEM images of $V_d-V_2O_3$ electrodes at different voltages in 30th circle.

Supplementary Figure 30 | SEM images of $V_d-V_2O_3$ electrodes at different voltages in 500th cycle.

(2) **Surface electronic states of $V_d-V_2O_3$ electrode:** The valence states of Zn and V on the surface of the $V_d-V_2O_3$ electrode changed during the charge/discharge process. The reversible insertion and extraction of Zn^{2+} are shown in **Supplementary Fig. 31a**. At a fully discharged state, the $V_d-V_2O_3$ electrode displays two Zn $2p_{3/2}$ components located at 1022.5 eV and 1023.2 eV which belong to the intercalated Zn^{2+} at different occupation sites (vanadium vacancies and tunnels nearby vanadium vacancies). At a fully charged state, the Zn $2p_{3/2}$ peak located at 1023.2 eV disappears while the peak of 1022.5 eV is reserved. That is to say that some vanadium vacancies occupied by Zn^{2+} are riveted in the lattice of $V_d-V_2O_3$, only enable Zn^{2+} reversibly (de)intercalation in the tunnel neighboring the remaining vanadium vacancies. Furthermore, the valence changes in the whole cycle are also shown in the high-resolution XPS of V (**Supplementary Fig. 31b**) where the peak of V^{4+} becomes dominant upon charging while releasing a sign of a let-up upon discharging.

Supplementary Figure 31 | Ex-situ XPS of $V_d-V_2O_3$ electrodes. (a) XPS high resolution spectrum of Zn 2p region. (b) XPS high resolution spectrum of V 2p region.

(3) **Phase of $V_d-V_2O_3$ electrode:** We tested the phase of the $V_d-V_2O_3$ electrode with different numbers of cycles during the charge/discharge process by XRD (Supplementary Fig. 27). It is found that even after several hundred cycles, the material structure on the surface of the $V_d-V_2O_3$ electrode remains unchanged, which further indicates the stability of the $V_d-V_2O_3$ electrode during the charge/discharge process.

Supplementary Figure 27 | XRD patterns of $V_d-V_2O_3$ electrodes under different cycles.

8. Authors claim two-step (de)intercalation of Zn^{2+} , what are those two sites that promote (de)intercalation at two different potentials.?

Reply: Thanks. By comparing the CV curves of the $\text{V}_d\text{-V}_2\text{O}_3$ electrode and the $\text{c-V}_2\text{O}_3$ electrode, we found that the redox peaks position was basically the same (**Fig. R3**). Consequently, the position of the redox peaks in CV curves is not affected by vanadium vacancies. These two redox peaks correspond to two steps of Zn^{2+} (de)intercalation from the host material [*Angew. Chem. Int. Ed.*, 2018, 57, 3943-3948; *Energy Environ. Sci.*, 2019, 12, 2273-2285; *Adv. Mater.*, 2018, 30, 1705580], which will bring out redox reactions, interrelating to the change of V valence state. Among the two redox peaks of the $\text{V}_d\text{-V}_2\text{O}_3$ electrode, the peak pair residing at higher voltage (1 V) is related to the redox pair of $\text{V}^{5+}/\text{V}^{4+}$ and the transition of the $\text{V}^{4+}/\text{V}^{3+}$ pair appears around 0.7V [*Nat. Commun.*, 2018, 9, 1656; *Sci. Adv.*, 2019, 5, eaax4279; *Adv. Funct. Mater.*, 2019, 30, 1907684].

Fig. R3 | a, CV curves of the 1st, 2nd, and 3rd cycles for $\text{V}_d\text{-V}_2\text{O}_3$ electrode at a scan rate of 0.1mV s^{-1} . **b**, CV curves of the 1st, 2nd, and 3rd cycles for $\text{c-V}_2\text{O}_3$ electrode at a scan rate of 0.1mV s^{-1} .

9. In Fig. 2, it is not clear, what the blue dots and red dots correspond to? Which one is for coulombic efficiency and which one represents capacity?

Reply: Thank you very much for your reminder.

In **Figure 2a** and **Figure 2b**, there is indeed an ambiguity between the blue dot and the red dot. The black dot is the charge capacity, while the red dot is the discharge

capacity. Furthermore, the blue Pentagram represents coulomb efficiency, and black and red dots correspond to capacity. Accordingly, we have made changes according to your suggestions, and then changed in **Figure 2a** and **Figure 2b** are as follows.

Fig. 2 | a, Rate performance at different current densities of V_d-V₂O₃ electrode. **b**, Long-term cycling performance and coulombic efficiency of V_d-V₂O₃ electrode at a current density of 5 A g⁻¹.

We have revised it in the manuscript accordingly.

10. The maximum specific capacity obtained is $\sim 187 \text{ mAh g}^{-1}$ at 0.1 A g^{-1} . The rate and capacity values are bit low, comparable to the many reports in the literature.

Reply: Thanks for pointing out this issue.

(1) The statement that the capacity values of V_d-V₂O₃ are a bit low, comparable to the many reports in the literature can be divided into two parts, namely the V₂O₃-based electrodes and the vanadium-based oxide electrodes.

① The results for the V_d-V₂O₃ show a bit lower capacity than other V₂O₃-based electrodes mainly attributes to the following three points.

First of all, the voltage window is wider than the voltage window in our manuscript, resulting in higher capacity. For example, the voltage window of the references (*ACS Nano* 2020, 14, 6, 7328-7337; *Small Methods* 2021, 2100578) is measured within the range of 0.2-1.6 V. Accordingly, as a comparison, we expanded the test voltage to 1.6 V and found that charging curves similar to that in the references appeared during the first cycle of charging. After the first cycle of charging, the V₂O₃ capacity increased sharply from 187 mA h g^{-1} at 0.1 A g^{-1} to 422 mA h g^{-1} (**Fig. R4a**). Furthermore, the CV

curves also arose similar oxidation peaks at the first cycle of charging to that in the references (**Fig. R4b**). Nevertheless, the oxidation of V_d - V_2O_3 at 1.6 V voltage will bring about structural changes (**Fig. R4c**), and the vanish of vanadium vacancies, which cannot provide excellent stability. After less than 1800 cycles, the capacity retention rate is only 86.2% (**Fig. R4d**). The reference (*ACS Nano* 2020, 14, 6, 7328-7337) also compared the charging to 1.3 V and 1.6 V respectively, and it is found that the capacity does change greatly. Consequently, we did not expand the voltage window to 1.6 V. Secondly, the electrode material undergoes a phase change during the cycle. For instance, the electrode material is converted from V_2O_3 material to $V_2O_{5-x} \cdot nH_2O$ after the first cycle of anodic oxidation in the reference (*ACS Nano* 2020, 14, 6, 7328-7337), which is actually the capacity of $V_2O_{5-x} \cdot nH_2O$. Furthermore, the V_2O_3 phase disappears after the initial charge process, and $Zn_{3+x}(OH)_{2+3x}V_{2-x}O_{7-3x} \cdot 2H_2O$ and zinc vanadate (Zn_yVO_z) phases arise from the second cycle in the reference (*Small Methods* 2021, 2100578). However, the structure of our V_2O_3 electrode has not changed after many cycles (**Supplementary Fig. 27**). Last but not least, the reference (*ACS Nano* 2020, 14, 6, 7328-7337) stated that carbon materials are unsuitable for the storage of Zn^{2+} , and the carbon content was excluded in the calculation of the capacity. Nevertheless, we include the carbon content in the calculation, which will also cause the V_d - V_2O_3 electrode to have a lower capacity than the reference.

② Compared with many other vanadium-based oxide electrodes [*Sci. Adv.*, 2019, 5, eaax4279; *Energy Environ. Sci.*, 2021, 14, 4095-4106; *Energy Environ. Sci.*, 2019, 12, 2273-2285; *Angew. Chem. Int. Ed.*, 2019, 58, 7062-7067; *Adv. Mater.*, 2018, 30, 1703725; *Adv. Mater.*, 2018, 30, 1705580], the voltage window is wider than the V_d - V_2O_3 electrode bringing about higher capacity. Moreover, the voltage of these vanadium-based oxide electrodes is not less than 1.6 V, while the V_d - V_2O_3 electrode will oxidize at 1.6 V, resulting in the disappearance of vanadium vacancies and structural changes, which will affect its ultra-long cycle stability.

Fig. R4 | **a**, Rate performance at different current densities of V_d-V₂O₃ electrode in 0.1-1.6 V voltage window. **b**, CV curves of V₂O₃ in Zn(CF₃SO₃)₂ electrolyte at 0.1 mV s⁻¹. **c**, XRD pattern of the V_d-V₂O₃ electrode charged to 1.6 V after 10 cycles. **d**, Long-term cycling performance and coulombic efficiency of V_d-V₂O₃ electrode at a current density of 5 A g⁻¹ in 0.1-1.6 V voltage window.

Supplementary Figure 27 | XRD patterns of V_d-V₂O₃ electrodes under different cycles.

(2) Although V_d-V₂O₃ has a higher diffusion coefficient of Zn²⁺, its conductivity

is lower ($832 \Omega \text{ cm}$ at 300K), bringing out the mediocre rate capability of the electrode.

However, the rate performance of the $\text{V}_d\text{-V}_2\text{O}_3$ electrode is still more remarkable than that of many electrode materials [*Energy Environ. Sci.*, 2021, 14, 3954-3964; *Adv. Energy Mater.*, 2019, 9, 1803815; *Nano Lett.*, 2018, 18, 1758-1763; *Adv. Funct. Mater.*, 2020, 30, 1907684; *Small Methods*, 2020, 4, 1900670] (**Supplementary Fig. 19 and Supplementary Table 4**).

Supplementary Figure 19 | Comparison of the capacity retention between $\text{V}_d\text{-V}_2\text{O}_3$ and other vanadium-based cathodes for aqueous ZIBs.

Supplementary Table 4 | Comparison of capacity retention for different vanadium-based aqueous ZIBs.

Sample	Current Density (A g^{-1})	Capacity Retention Ratio(%)	Ref.
MnVO	0.2-2	50.0	Energy Environ. Sci. , 2021, 14, 3954-3964
$\text{O}_d\text{-MnO}_2$	0.2-2	51.4	Adv. Energy Mater. , 2019, 9, 1803815
$\text{Na}_2\text{V}_6\text{O}_{16}\cdot 1.63\text{H}_2\text{O}$	0.1-1	60.6	Nano Lett. , 2018, 18, 1758-1763
$\text{Mn}_{0.15}\text{V}_2\text{O}_5\cdot n\text{H}_2\text{O}$	0.5-5	66.7	Adv. Funct. Mater. , 2020, 30, 1907684
C-KVO	0.2-2	68.9	Small Methods , 2020, 4, 1900670
$\text{V}_d\text{-V}_2\text{O}_3$	0.1-1	70.4	This work

11. According to the authors, the calculated b values for both cathode and anode peaks from CV curves are 0.94, 0.75, 0.79 and 0.88, respectively. This suggests the Faradaic

component of charging and discharging are not equal, then how could authors cycle without issues for long.?

Reply: Thanks for the useful suggestion.

For the b value, it can be used to qualitatively distinguish between pseudocapacitance and battery-type materials and provide more kinetic information of electrochemical reactions, including charge storage types at different potentials/scan rates and charge storage mechanisms of different ion intercalation batteries [*Nat. Energy*, 2017, 2, 17105; *Adv. Sci.*, 2017, 5, 1700322]. Besides, most of the previous literatures, including this manuscript, uses formula 1 to calculate the contribution of pseudocapacitance in cyclic voltammetry curves at present.

$$i = av^b \quad (1a)$$

$$i(V) = k_1v + k_2v^{1/2} \quad (1b)$$

Nevertheless, there are many ambiguities and even misunderstandings as listed below in this method.

- 1) The influence of overvoltage caused by resistance on voltage is not considered [*Nat. Mater.*, 2010, 9, 146-151; *Adv. Mater.*, 2017, 29, 1605535];
- 2) The residual current when the scanning direction is reversed is not corrected, which may bring about the deviation between the fitting curve and the redox peak in the experimental curve, and even the illusion that the fitting value exceeds the experimental value [*Nano Energy*, 2020, 77, 105069];
- 3) The double-layer capacitance current is not separated from the pseudocapacitance current, resulting in the over-exaggeration of pseudocapacitance [*Energy Environ. Mater.*, 2019, 2, 30-37].

Accordingly, the b value can only qualitatively distinguish between pseudocapacitive materials and battery-type materials and supply electrochemical kinetics information but is unsuited for qualitative comparison. Even more remarkable, under the same sweep rate and different voltages, the b values are constantly changing [*J. Am. Chem. Soc.*, 2011, 133, 16291-16299], which further indicates that only the b value at some points cannot be used to determine the stability of the battery material. Moreover, some references have unequal b values at the redox peaks, but they still have excellent

stability [*Sci. Adv.*, 2019, 5, eaax4279; *Adv. Funct. Mater.*, 2019, 30, 1907684; *ACS Nano*, 2020, 14, 7328-7337]. It can be concluded from the above discussion that the cycle without issues for long cannot be judged only by the Faradaic component of charging and discharging. Accordingly, we have mentioned this point in the revised manuscript.

12. With the increase in scan rate, the capacitive contribution increasing (Fig. 2f), if so what is the use of having V_d -vacancy in the bulk and especially for high rate performance.?

Reply: Thanks for the useful suggestion.

As the scanning rate increases, the capacitive contribution increases. Even so, the vanadium vacancies in the bulk for high rate performance still plays a role in stabilizing the material structure and effectively weakening the strong interaction between Zn^{2+} and V_2O_3 material host. The detailed proofs are as follows.

To compare the capacitance contribution of c- V_2O_3 and V_d - V_2O_3 electrodes at a high rate, we quantified the contribution of diffusion-controlled and capacitive-controlled at different scan rates (**Figs. R5a and R5b**). With the increase of scan rates from 0.2 to 1.0 $mV s^{-1}$, the capacitance contribution rates of the c- V_2O_3 electrode increase from 74.2% to 86.8%. At the same scan rate, the c- V_2O_3 electrode has more capacitance contribution than the V_d - V_2O_3 electrode. However, the cycle stability of the c- V_2O_3 electrode is far inferior to that of the V_d - V_2O_3 electrode, which indicates that the high ratio of contribution of pseudocapacitance does not mean that it has excellent cycle stability at a high rate. On the contrary, the V_d - V_2O_3 electrode has more diffusion control contribution in the bulk than the c- V_2O_3 electrode at the same scanning rates. During the diffusion process, it is precisely because part of vanadium vacancies in the bulk provides permanent sites for the preoccupation of a small amount of Zn^{2+} so that the system will have a more stable structure to against collapsing during the process of Zn^{2+} insertion/extraction. Meanwhile, the other vanadium vacancies can effectively weaken the strong interaction between Zn^{2+} and the V_2O_3 material host to allow free insertion/extraction of Zn^{2+} . Consequently, it is believing that the vanadium vacancies

in the bulk can still play a role in stabilizing the material structure and effectively weakening the strong interaction between Zn^{2+} and V_2O_3 material host at a high rate.

Fig. R5 | **a**, Contribution ratios of capacitive and diffusion-controlled capacities at different scan rates from 0.2 to 1.0 mV s^{-1} for c- V_2O_3 electrode. **b**, Contribution ratios of capacitive and diffusion-controlled capacities at different scan rates from 0.1 to 1.0 mV s^{-1} for V_d - V_2O_3 electrode.

13. In supplementary Fig. 17, why the $D_{\text{Zn}^{2+}}$ values do not follow a trend? Give complete details of the calculation involved in estimating $D_{\text{Zn}^{2+}}$. (parameters and their values).

Reply: Thanks for the useful suggestion.

The $D_{\text{Zn}^{2+}}$ values have a downward trend as a whole during the charging/discharging process, but there will drop more sharply at 1 V (charging process), 0.5 V (discharging process), and fully discharged (**Supplementary Fig. 24**). The detailed discussion is as follows (Supplementary Fig. 17 has been revised to Supplementary Fig. 24).

The decrease of $D_{\text{Zn}^{2+}}$ values mainly arises at the platform of charging and discharging. The charge/discharge plateau of the battery is interrelated to the redox peak in the CV curve. When charging (discharging), the decrease plateau of $D_{\text{Zn}^{2+}}$ value is about 1 V (0.5 V), corresponding to peak 2 (peak 3) in **Fig. R6c**. By calculating the b value of CV peak, the related b values of peak 2 and peak 3 are 0.75 and 0.79 respectively (**Fig. R6d**). This indicates that the storage behavior of Zn^{2+} at this voltage

is mainly controlled by both ion diffusion and capacitance control, and Zn^{2+} at these two voltages is more dominated by ion diffusion than peak 1 ($b = 0.94$) and peak 4 ($b = 0.88$). Owing to the lattice of $V_d-V_2O_3$ shrinks (Figs. R6a and R6b) and the influence of ion diffusion, the diffusion rate decreases naturally at these two voltages. When fully discharged, the lattice shrinkage of the host material reaches the maximum, and it is difficult for Zn^{2+} to diffuse in $V_d-V_2O_3$ bringing out the decrease in the value of $D_{Zn^{2+}}$ (Nat. Energy, 2021, 6, 706-714).

Supplementary Figure 24 | GITT curves and calculation of corresponding Zn^{2+} diffusion coefficient of $V_d-V_2O_3$ electrode.

Fig. R6 | **a**, *Ex-situ* XRD patterns of $V_d-V_2O_3$ electrodes at different cut-off voltages during the charge and discharge process. **b**, An enlarged view of the red dotted frame in Figure 4a. **c**, CV curves of the $V_d-V_2O_3$ electrode at scan rates ranging from 0.1 to 1 $mV s^{-1}$. **d**, Log (i) versus log (v) plots at specific peak currents.

The complete details of the calculation involved in estimating $D_{Zn^{2+}}$ values are set out below.

The Galvanostatic Intermittent Titration Technique (GITT) was used to analyze the reaction and diffusion kinetics at 150 mA g^{-1} current density, 1 min charge/discharge time, and 30 min standing time. In the whole process of charge and discharge, the program is repeatedly applied to the battery. The diffusion coefficient of Zn^{2+} ($D_{Zn^{2+}}$, $cm^2 s^{-1}$) was calculated by GITT and based on the following formula:

$$D_{Zn} = \frac{4}{\pi\tau} \left(\frac{m_B V_m}{M_B A} \right)^2 \left(\frac{\Delta E_S}{\Delta E_\tau} \right)^2$$

Where τ represents the duration of the current pulse, m_B corresponds to the mass of the active material, M_B and V_B are related to the molecular mass ($g mol^{-1}$) and molar

volume($\text{cm}^3 \text{ mol}^{-1}$), respectively. A is the total volume of the electrode in contact with the electrolyte, ΔE_s is the open-circuit voltage (V) difference measured at the end of two successive relaxation cycles, and ΔE_t is the voltage change (V) during the constant current pulse (**Supplementary Fig. 25**).

Supplementary Figure 25 | Schematic illustration of a selected single step of the GITT profile during charging.

We have added these data and additional discussions in our revised Supplementary Information accordingly.

REVIEWERS' COMMENTS

Reviewer #1 (Remarks to the Author):

The manuscript is now well revised according to the reviewer's comments and suggestions. It is now acceptable to publish in Nature Communications.

Reviewer #2 (Remarks to the Author):

The reviewer finds that all the comments pointed out by the reviewers have been considered by the authors in the revised version.

The reviewer recommends accepting the paper in its actual form in nature communications.

Reviewer #3 (Remarks to the Author):

The authors have addressed all the comments raised satisfactorily.